# On Architectural Compression of Text-to-Image Diffusion Models

## Abstract

Exceptional text-to-image (T2I) generation results of Stable Diffusion models (SDMs) come with substantial computational demands. To resolve this issue, recent research on efficient SDMs has prioritized reducing the number of sampling steps and utilizing network quantization. Orthogonal to these directions, this study highlights the power of classical architectural compression for general-purpose T2I synthesis by introducing a block-removed knowledge-distilled SDM (BK-SDM). We eliminate several residual and attention blocks from the U-Net of SDMs, obtaining over a 30% reduction in the number of parameters, MACs per sampling step, and latency. We conduct distillation-based pretraining with only 0.22M LAION pairs (fewer than 0.1% of the full training pairs) on a single A100 GPU. Despite being trained with limited resources, our compact models can imitate the original SDM by benefiting from transferred knowledge and achieve competitive results against larger multi-billion parameter models on the zero-shot MS-COCO benchmark. Moreover, we demonstrate the applicability of our lightweight pretrained models in personalized generation with DreamBooth finetuning.

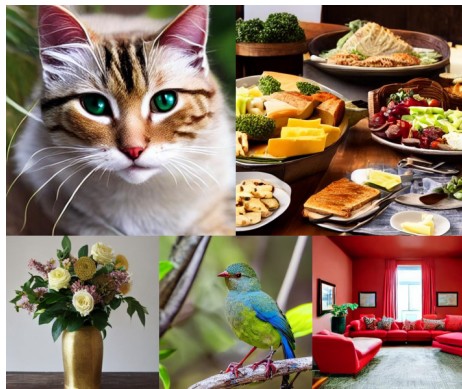

**(a) Efficient General-purpose T2I**

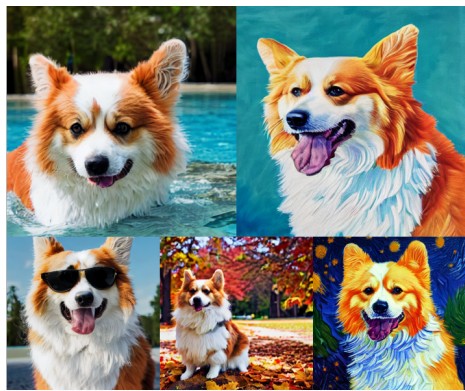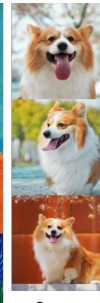

Input
Subject

**(b) Efficient Personalized T2I**

Figure 1: Our compressed stable diffusion enables efficient (a) zero-shot general-purpose text-to-image generation and (b) personalized synthesis. Selected samples from our lightest BK-SDM-Small with 36% reduced parameters and latency are shown.

Submitted to 37th Conference on Neural Information Processing Systems (NeurIPS 2023). Do not distribute.

# 1 Introduction

Large diffusion models [44, 51, 38, 47] have showcased groundbreaking results in text-to-image (T2I) synthesis tasks, which aim to create photorealistic images from textual descriptions. Stable Diffusion models (SDMs) [46, 47] are one of the most renowned open-source models, and their exceptional capability has begun to be leveraged as a backbone in several text-guided vision applications, e.g., text-driven image editing [2, 23] and 3D object creation [67], text-to-video generation [1, 68], and subject-driven [50, 25] and controllable [37, 71] T2I.

SDMs are T2I-specialized latent diffusion models (LDMs) [47], which employ diffusion operations [17, 59, 30] in a latent space to improve compute efficiency. Within a SDM, a U-Net [49, 6] conducts an iterative sampling procedure to gradually eliminate noise from random latents and is assisted by a text encoder [42] and an image decoder [9, 64] to produce text-aligned images. This inference process still involves excessive computational requirements (see Figure 2), which often hinder the utilization of SDMs despite their rapidly growing usage.

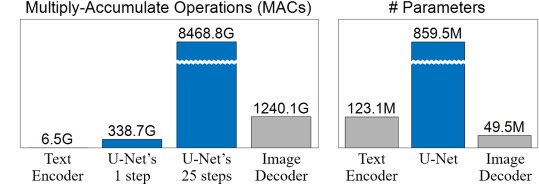

Figure 2: Computation of the major components in Stable Diffusion v1. The denoising U-Net is the main processing bottleneck. THOP [75] is used to measure MACs in generating a 512×512 image.

To alleviate this issue, numerous approaches toward efficient SDMs have been introduced. Meng et al. [35, 34] reduce the number of denoising steps by distilling a pretrained diffusion model to guide an identically architectured model with fewer sampling steps. Li et al. [28], Hou and Asghar [19], Shen et al. [57] employ post-training quantization techniques, and Chen et al. [4] enhance the implementation of SDMs for better compatibility with GPUs. However, the removal of architectural elements in diffusion models has not been investigated in spite of the established efficacy of structured pruning across discriminative models [26, 69] and generative adversarial networks (GANs) [31, 24].

This study unlocks the immense potential of classical architectural compression in attaining smaller and faster diffusion models. We eliminate multiple residual and attention blocks from the U-Net of a SDM and pretrain it with feature-level knowledge distillation (KD) [48, 13] for general-purpose T2I synthesis. Despite being trained with only 0.22M LAION pairs (less than 0.1% of the entire training pairs) [55] on a single A100 GPU, our compact models can mimic the original SDM by leveraging transferred knowledge. On the popular zero-shot MS-COCO benchmark [29], our work achieves a FID [15] score of 15.76 with 0.76B parameters and 16.98 with 0.66B parameters, which are on par with multi-billion parameter models [43, 7, 8]. Furthermore, we present the practical application of our lightweight pretrained models in customized T2I with DreamBooth finetuning [50].

Our contributions are summarized as follows:

- To the best of our knowledge, this is the first study to architecturally compress large-scale diffusion models. Our work is orthogonal to prior directions for efficient diffusion, e.g., enabling less sampling steps and employing quantization, and can be readily integrated with them.
- We compress SDMs by removing architectural blocks from the U-Net and achieve more than 30% reduction in model size and inference speed. We also introduce an interesting finding on the minor role of innermost blocks.
- We demonstrate the advantage of distillation-based pretraining, which allows us to attain competitive zero-shot T2I results even with very limited training resources.
- We highlight the capability of our light pretrained backbones in customized generation. Our models can lower the finetuning cost by 30% while retaining 97% scores of the original SDM.

# 2 Related work

**Large T2I diffusion models.** By gradually removing noise from corrupted data, diffusion-based generative models [18, 59, 6] enable high-fidelity synthesis with broad mode coverage. Integrating these merits with the advancement of pretrained language models [42, 41, 5] has significantly improved the quality of T2I synthesis. In GLIDE [38] and Imagen [51], a text-conditional diffusion

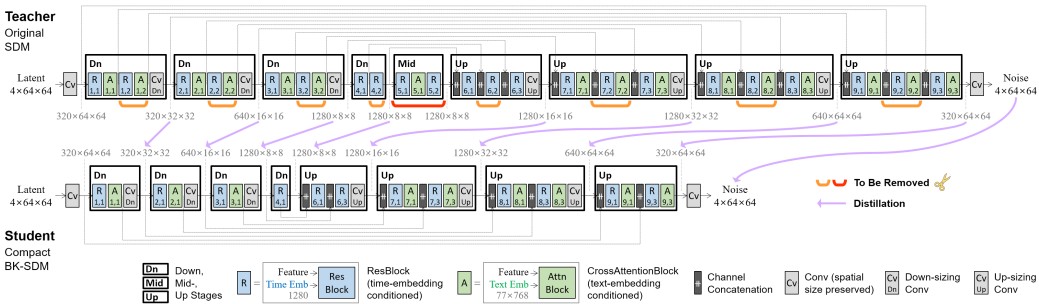

Figure 3: U-Net architectures of SDMs and KD-based pretraining process. The compact U-Net student is built by eliminating several residual and attention blocks from the original U-Net teacher. Through the feature and output distillation from the teacher, the student can be trained effectively yet rapidly. See Appendix for the details of block components.

model generates a 64×64 image, which is upsampled via super-resolution modules. In DALL·E-2 [44], a text-conditional prior network produces an image embedding, which is transformed into a 64×64 image via a diffusion decoder and further upscaled into higher resolutions. SDMs [46, 47] perform the diffusion modeling in a 64×64 latent space constructed through a pixel-space autoencoder. We use SDM as our baseline because of its open-access and gaining popularity over numerous downstream tasks [2, 67, 1, 50].

**Efficient diffusion models.** Several studies have addressed the slow sampling process of diffusion models. Diffusion-tailored distillation approaches [35, 34, 52] progressively transfer knowledge from a pretrained diffusion model to a fewer-step model with the same architecture. Fast high-order solvers [32, 33, 73] for diffusion ordinary differential equations boost the sampling speed. Orthogonal to these directions for less sampling steps, our network compression approach reduces per-step computation and can be easily integrated with them. Leveraging quantization techniques [28, 19, 57] and implementation optimizations [4] has been applied for SDMs and also can be combined with our models for further efficiency gains.

**Distillation-based compression.** KD enhances the performance of small-size models by exploiting output-level [16, 39] and feature-level [48, 13, 70] information of large source models. Although this classical distillation has been actively used toward efficient GANs [27, 45, 31, 22, 72], its power has not been explored for structurally compressed diffusion models. Distillation-based pretraining enables small yet capable general-purpose language models [54, 61, 21] and vision transformers [63, 11]. Beyond such models, we show that its success can be extended to diffusion models with iterative sampling steps. Concurrently with our study, a recently released small SDM without paper evidence [40] similarly utilizes KD pretraining for a block-eliminated architecture, but it relies on significantly more training resources along with multi-stage distillation. In contrast, our lightest model achieves further reduced computation, and we show that competitive results can be obtained even with much less data and single-stage distillation.

## 3 BK-SDM: block-removed knowledge-distilled SDM

We compress the U-Net [49] of a SDM [46, 47], which is the most compute-heavy component (see Figure 2). Conditioned on the text and time-step embeddings, the U-Net performs multiple denoising steps on latent representations. At each denoising step, the U-Net produces the noise residual to compute the latent for the next step (see the top part of Figure 3). We reduce this per-step computation by exploiting block-level elimination and feature distillation.

### 3.1 Compressed U-Net architecture

The proposed models are referred to as:

- ○ BK-SDM-Base (0.76B parameters) obtained with Section 3.1.1 (fewer blocks in outer stages).
- ○ BK-SDM-Small (0.66B) with Section 3.1.1 (fewer blocks) and Section 3.1.2 (mid-stage removal).

Table 1: Minor impact of eliminating the mid-stage from the U-Net of SDM on zero-shot MS-COCO performance. Any retraining is not performed for the mid-stage removed model. For evaluation details, see Section 5.1.1.

| Model | Performance | | # Params | |
|---|---|---|---|---|
| | FID ↓ | IS ↑ | U-Net | Whole |
| SDM-v1.4 [46] | 13.05 | 36.76 | 859.5M | 1032.1M |
| Mid-Stage Removal | 15.60 | 32.33 | 762.5M (-11.3%) | 935.1M (-9.4%) |

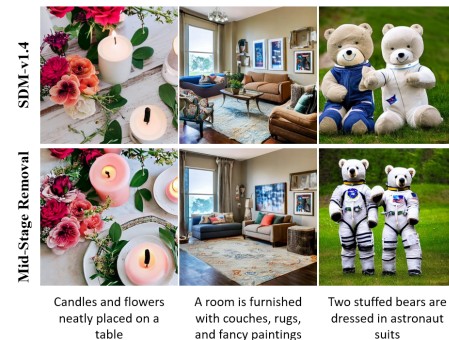

Candles and flowers neatly placed on a table

A room is furnished with couches, rugs, and fancy paintings

Two stuffed bears are dressed in astronaut suits

Figure 4: Visual results of the mid-stage removed U-Net without retraining.

### 3.1.1 Fewer blocks in the down and up stages

Our design philosophy is closely aligned with that of DistilBERT [54] which halves the number of layers for improved computational efficiency and initializes the compact model with the original weights by benefiting from the shared dimensionality. In the original U-Net, each stage with a common spatial size consists of multiple blocks, and most stages contain pairs of residual (R) [12] and cross-attention (A) [65, 20] blocks. We hypothesize the existence of some unnecessary pairs and use the following removal strategies, as shown in Figure 3.

For the down stages, we maintain the first R-A pairs while eliminating the second pairs, because the first pairs process the changed spatial information and would be more important than the second pairs. This design choice does not harm the dimensionality of the original U-Net, enabling the use of the corresponding pretrained weights for initialization [54].

For the up stages, while adhering to the aforementioned scheme, we retain the third R-A pairs. This allows us to utilize the output feature maps at the end of each down stage and the corresponding skip connections between the down and up stages. The same process is applied to the innermost down and up stages that contain only R blocks.

### 3.1.2 Removal of the entire mid-stage

Surprisingly, removing the entire mid-stage from the original U-Net (marked with red in Figure 3) does not noticeably degrade the generation quality for many text prompts while effectively reducing the number of parameters (see Table 1 and Figure 4). This observation is consistent with the minor role of inner layers in the U-Net generator of GANs [24].

Integrating the mid-stage removal with fewer blocks in Section 3.1.1 further decreases computational burdens (Table 3) at the cost of a slight decline in performance (Table 2). Therefore, we offer this mid-stage elimination as an option, depending on the priority between compute efficiency and generation quality.

## 3.2 Distillation-based pretraining

For general-purpose T2I generation, we train the compact U-Net to mimic the behavior of the original U-Net. Following Rombach et al. [47], we use the pretrained-and-frozen encoders to obtain the inputs of the U-Net.

Given the latent representation $z$ of an image and its paired text embedding $y$, the task loss for the reverse denoising process [18, 47] is computed as:

$$\mathcal{L}_{\text{Task}} = \mathbb{E}_{z,\epsilon,y,t}\Big[||\epsilon - \epsilon_{\text{S}}(z_t, y, t)||_2^2\Big], \tag{1}$$

where $\epsilon \sim N(0, I)$ and $t \sim \text{Uniform}(1, T)$ denote the noise and time step sampled from the diffusion process, respectively, and $\epsilon_{\text{S}}(\circ)$ indicates the output of our compact U-Net student. For brevity, we omit the subscripts of $\mathbb{E}_{z,\epsilon,y,t}[\circ]$ in the following notations.

The compact student is also trained to imitate the outputs of the original U-Net teacher, $\epsilon_T(\circ)$ ,with the following output-level KD objective [16]:

$$\mathcal{L}_{\text{OutKD}} = \mathbb{E}\Big[||\epsilon_T(z_t, y, t) - \epsilon_S(z_t, y, t)||_2^2\Big]. \tag{2}$$

A key to our approach is the utilization of feature-level KD [48, 13] that provides abundant guidance for the student's training:

$$\mathcal{L}_{\text{FeatKD}} = \mathbb{E}\Big[\sum_l ||f_T^l(z_t, y, t) - f_S^l(z_t, y, t)||_2^2\Big], \tag{3}$$

where $f_T^l(\circ)$ and $f_S^l(\circ)$ represent the feature maps of the $l$-th layer in a predefined set of distilled layers from the teacher and the student, respectively. While learnable regressors (e.g., 1×1 convolutions to match the number of channels) have been commonly used in existing studies [58, 45, 48], our approach circumvents this requirement. By applying distillation at the end of each stage in both models, we ensure that the dimensionality of the feature maps already matches, thus eliminating the need for additional regressors.

The final objective is formalized as below, and we simply set the loss weights $\lambda_{\text{OutKD}}$ and $\lambda_{\text{FeatKD}}$ as 1. Without any hyperparameter tuning, our approach is effective in empirical validation.

$$\mathcal{L} = \mathcal{L}_{\text{Task}} + \lambda_{\text{OutKD}}\mathcal{L}_{\text{OutKD}} + \lambda_{\text{FeatKD}}\mathcal{L}_{\text{FeatKD}}. \tag{4}$$

### 3.3 Application: faster and smaller personalized SDMs

To emphasize the benefit of our lightweight pretrained SDMs, we use a popular finetuning scenario for personalized generation. DreamBooth [50] enables T2I diffusion models to create contents about a particular subject using just a few input images. Our compact models not only accelerate inference speed but also reduce finetuning cost. Moreover, they produce high-quality images based on the inherited capability of the original SDM.

## 4 Experimental setup

### 4.1 Datasets and evaluation metrics

**Pretraining.** We train our compact SDM with only 0.22M image-text pairs from LAION-Aesthetics V2 6.5+ [55, 56], which are significantly fewer than the original training data used for SDM-v1.4 [46] (i.e., 600M pairs of LAION-Aesthetics V2 5+ [55] for the resumed training).

**Zero-shot T2I evaluation.** Following the popular protocol [43, 47, 51] to assess general-purpose T2I with pretrained models, we use 30K prompts from the MS-COCO validation split [29] and compare the generated images to the whole validation set. We compute Fréchet Inception Distance (FID) [15] and Inception Score (IS) [53] to assess visual quality. Moreover, we measure CLIP score [42, 14] with CLIP-ViT-g/14 model to assess text-image correspondence.

**Finetuning for personalized generation.** We use the DreamBooth dataset [50] that covers 30 subjects, each of which is associated with 25 prompts and 4∼6 images. Through individual finetuning for each subject, 30 personalized models are obtained. For evaluation, we follow the protocol of Ruiz et al. [50] based on four synthesized images per subject and per prompt. We consider CLIP-I and DINO scores to measure how well subject details are maintained in generated images (i.e., subject fidelity) and CLIP-T scores to measure text-image alignment (i.e., text fidelity). We use ViT-S/16 embeddings [3] for DINO scores and CLIP-ViT-g/14 embeddings for CLIP-I and CLIP-T.

### 4.2 Implementation

We use the released version v1.4 of SDM [46] as our compression target. We remark that our approach is also applicable to other versions in v1.1–v1.5 with the same architecture and to SDM-v2 with a similarly designed architecture.

Table 2: Zero-shot results on 30K prompts from MS-COCO validation set [29] at 256×256 resolution. Despite being trained with a smaller dataset and having fewer parameters, our compressed models achieve results on par with prior approaches for general-purpose T2I. For our models, the results with the minimum FID and the final 50K-th iteration are reported (see Section 5.1.3 for detailed analysis).

| Model | Type | FID ↓ | IS ↑ | # Params | Data Size |
|---|---|---|---|---|---|
| SDM-v1.4 [47] | DF | 13.05 | 36.76 | 1.04B | 600M |
| Small Stable Diffusion [40] | DF | 12.76 | 32.33 | 0.76B | 229M |
| BK-SDM-Base (Ours) @ Min FID | DF | 13.57 | 29.22 | 0.76B | 0.22M |
| BK-SDM-Base (Ours) @ Final Iter | DF | 15.76 | 33.79 | 0.76B | 0.22M |
| BK-SDM-Small (Ours) @ Min FID | DF | 15.93 | 29.61 | 0.66B | 0.22M |
| BK-SDM-Small (Ours) @ Final Iter | DF | 16.98 | 31.68 | 0.66B | 0.22M |
| DALL·E$^{\dagger\star}$ [43] | AR | 27.5 | 17.9 | 12B | 250M |
| CogView$^{\ddagger\star}$ [7] | AR | 27.1 | 18.2 | 4B | 30M |
| CogView2$^{\dagger\star}$ [8] | AR | 24.0 | 22.4 | 6B | 30M |
| Make-A-Scene$^{\ddagger}$ [10] | AR | 11.84 | - | 4B | 35M |
| LAFITE$^{\ddagger\sharp}$ [74] | GAN | 26.94 | 26.02 | 0.23B | 3M |
| GALIP (CC3M)$^{\dagger}$ [62] | GAN | 16.12 | - | 0.32B | 3M |
| GALIP (CC12M)$^{\dagger}$ [62] | GAN | 12.54 | - | 0.32B | 12M |
| GLIDE$^{\ddagger}$ [38] | DF | 12.24 | - | 5B | 250M |
| LDM-KL-8-G$^{\ddagger\sharp}$ [47] | DF | 12.63 | 30.29 | 1.45B | 400M |
| DALL·E-2$^{\dagger}$ [44] | DF | 10.39 | - | 5.2B | 250M |

$\dagger$ and $\ddagger$: FID from [62] and [47], respectively. $\star$ and $\sharp$: IS from [8] and [47], respectively. DF and AR: diffusion and autoregressive models. ↓ and ↑: lower and higher values are better.

We adjust the codes in Diffusers library [66] for pretraining our models and those in PEFT library [60] for DreamBooth-finetuning, both of which adopt the training process of DDPM [18] in latent spaces. We use a single NVIDIA A100 80G GPU for 50K-iteration pretraining with a constant learning rate of 5e-5. For DreamBooth, we use a single NVIDIA GeForce RTX 3090 GPU to finetune each personalized model for 800 iterations with a constant learning rate of 1e-6.

Following the default inference setup, we use PNDM scheduler [30] for zero-shot T2I generation and DPM-Solver [32, 33] for DreamBooth results. For compute efficiency, we always opt for 25 denoising steps of the U-Net at the inference phase. The classifier-free guidance scale [17, 51] is set to the default value of 7.5, except the analysis in Figure 7.

# 5 Results

## 5.1 General-purpose T2I generation

### 5.1.1 Main results

Table 2 shows the zero-shot T2I results on 30K samples from the MS-COCO 256×256 validation set. Despite being trained with only 0.22M samples and having fewer than 1B parameters, our compressed models demonstrate competitive performance on par with previous large pretrained models. Despite the absence of a paper support, we include the model [40] that is identical in structure to BK-SDM-Base for comparison. This model benefits from far more training resources, i.e., two-stage KD relied on two teachers (SDM-v1.4 and v1.5) and a much larger volume of data with significantly longer iterations.

Figure 5 depicts synthesized images of different models with some MS-COCO captions. Our compressed models inherit the superior ability of SDM and produce more photorealistic images compared to the AR-based [8] and GAN-based [74, 62] baselines. Noticeably, the same latent code results in a shared visual style between the original and our compact SDMs (4th–6th columns in Figure 5), similar to the observation in transfer learning for GANs [36].

Table 3 summarizes how the computational reduction for each sampling step of the U-Net impacts the overall compute of the entire SDM. The per-step reduction effectively decreases MACs and inference time by more than 30% as well as the number of parameters.

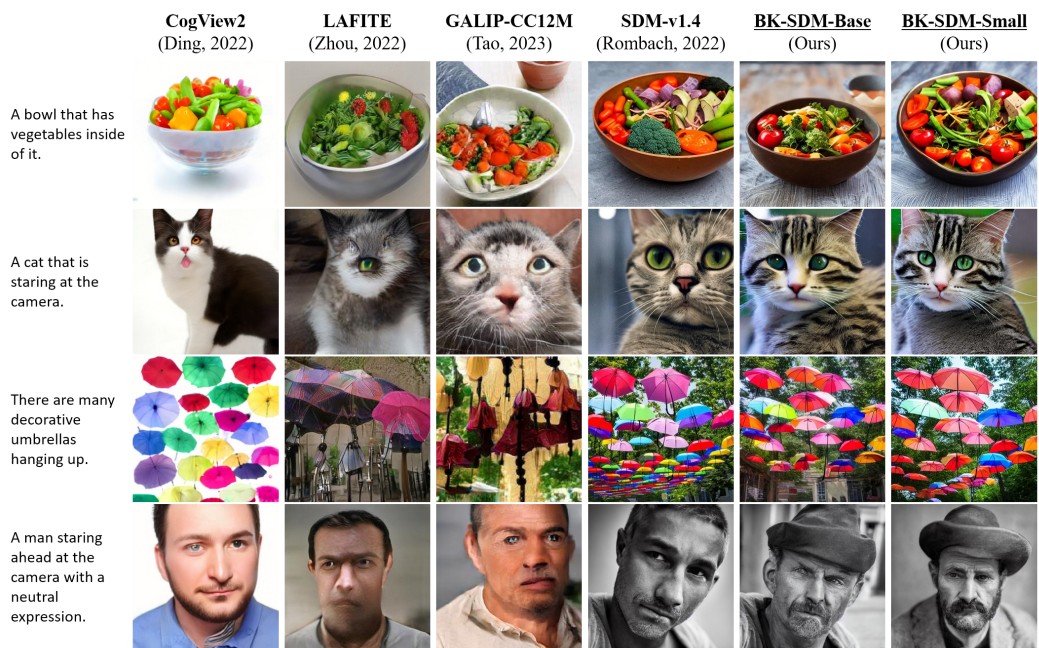

| CogView2 (Ding, 2022) | LAFITE (Zhou, 2022) | GALIP-CC12M (Tao, 2023) | SDM-v1.4 (Rombach, 2022) | BK-SDM-Base (Ours) | BK-SDM-Small (Ours) |

Figure 5: Visual comparison on zero-shot MS-COCO benchmark. The results of previous studies [8, 74, 62] were obtained with their official codes and released models. We do not apply any CLIP-based reranking for SDM and our models.

Table 3: The impact of per-step compute reduction of the U-Net on the entire SDM. The number of sampling steps is indicated with the parentheses, e.g., U-Net (1) for one step. The full computation (denoted by "Whole") covers the text encoder, U-Net, and image decoder. All corresponding values are obtained on the generation of a single 512×512 image with 25 denoising steps. The latency was measured on Xeon Silver 4210R CPU 2.40GHz and NVIDIA GeForce RTX 3090 GPU.

| Model | # Params | | MACs | | | CPU Latency | | | GPU Latency | | |
|---|---|---|---|---|---|---|---|---|---|---|---|
| | U-Net | Whole | U-Net (1) | U-Net (25) | Whole | U-Net (1) | U-Net (25) | Whole | U-Net (1) | U-Net (25) | Whole |
| SDM-v1.4 [46] | 860M | 1033M | 339G | 8469G | 9716G | 5.63s | 146.28s | 153.02s | 0.049s | 1.28s | 1.41s |
| BK-SDM-Base (Ours) | 580M (-32.6%) | 752M (-27.1%) | 224G (-33.9%) | 5594G (-33.9%) | 6841G (-29.5%) | 3.84s (-31.8%) | 99.95s (-31.7%) | 106.62s (-30.3%) | 0.032s (-34.6%) | 0.83s (-35.2%) | 0.96s (-31.9%) |
| BK-SDM-Small (Ours) | 483M (-43.9%) | 655M (-36.5%) | 218G (-35.7%) | 5444G (-35.7%) | 6690G (-31.1%) | 3.45s (-38.7%) | 89.78s (-38.6%) | 96.52s (-36.9%) | 0.030s (-38.7%) | 0.77s (-39.8%) | 0.90s (-36.1%) |

### 5.1.2 Ablation study

Table 4 presents the ablation study with the zero-shot MS-COCO benchmark dataset. The common default settings for the models N1–N7 involve the usage of fewer blocks in the down and up stages (Section 3.1.1) and the denoising task loss (Eq. 1). All the models are drawn at the 50K-th training iteration. We made the following observations.

**N1 *vs.* N2.** Importing the pretrained weights for initialization clearly improves the performance of block-removed SDMs. Transferring knowledge from well-trained models, a popularized practice in machine learning, is also beneficial for T2I generation with SDMs.

**N2 *vs.* N3 *vs.* N4.** Exploiting output-level KD (Eq. 2) effectively boosts the generation quality compared to using only the denoising task loss. Leveraging feature-level KD (Eq. 3) further improves the performance by offering sufficient guidance over multiple stages in the student.

**N4 *vs.* N5.** An increased batch size leads to a better IS and CLIP score but with a minor drop in FID. We opt for a batch size of 256 based on the premise that more samples per batch would enhance the model's understanding ability.

**N6 and N7.** Despite slight performance drop, the models N6 and N7 with the mid-stage removal have fewer parameters (0.66B) than N4 and N5 (0.76B), offering improved compute efficiency.

Table 4: Ablation study on zero-shot MS-COCO 256×256 30K. The common settings include fewer blocks in the down and up stages and the denoising task loss. N5 and N7 correspond to BK-SDM-Base and BK-SDM-Small, respectively

| | Model | | | | | Performance | | |
|---|---|---|---|---|---|---|---|---|
| No. | Initialize Weights | Output KD | Feature KD | Batch Size | Remove Mid | FID ↓ | IS ↑ | CLIP score ↑ |
| N1 | Random | ✗ | ✗ | 64 | ✗ | 43.80 | 13.61 | 0.1622 |
| N2 | Pretrained | ✗ | ✗ | 64 | ✗ | 20.45 | 22.68 | 0.2444 |
| N3 | Pretrained | ✓ | ✗ | 64 | ✗ | 16.48 | 27.30 | 0.2620 |
| N4 | Pretrained | ✓ | ✓ | 64 | ✗ | 14.61 | 31.44 | 0.2826 |
| N5 | Pretrained | ✓ | ✓ | 256 | ✗ | 15.76 | 33.79 | 0.2878 |
| N6 | Pretrained | ✓ | ✓ | 64 | ✓ | 16.87 | 29.51 | 0.2644 |
| N7 | Pretrained | ✓ | ✓ | 256 | ✓ | 16.98 | 31.68 | 0.2677 |
| Original SDM-v1.4 [46, 47] | | | | | | 13.05 | 36.76 | 0.2958 |

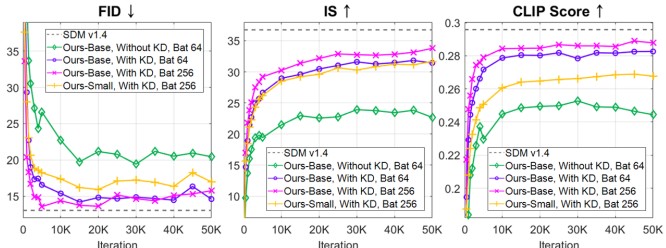

Figure 6: Results on zero-shot MS-COCO 256×256 30K over training progress. For our models, the architecture size, usage of KD, and batch size are denoted.

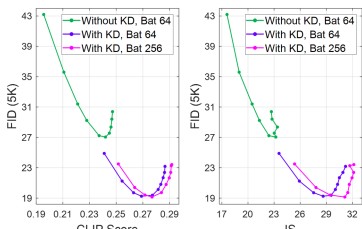

Figure 7: Effect of different classifier-free guidance scales on MS-COCO 512×512 5K.

### 5.1.3 Impact of distillation on pretraining phase

We further analyze the merits of transferred knowledge via distillation, with the models from the pretrained weight initialization. Figure 6 shows zero-shot T2I performance over training iterations. Compared to the absence of KD (indicated with green), distillation (purple and pink) accelerates the training process and leads to improved generation scores, demonstrating the benefits of providing sufficient hints for training guidance. Notably, our small-size model trained with KD (yellow) outperforms the bigger base-size model without KD (green). Additionally, while the best FID score is observed early on for our models, IS and CLIP score exhibit ongoing improvement, implying that judging models solely with FID may be suboptimal.

Figure 7 shows the trade-off curves from different classifier-free guidance scales [17, 51] $\{2.0, 2.5, 3.0, 3.5, 4.5, 5.5, 6.5, 7.5, 8.5, 9.5\}$. For the analysis, we use 5K samples from the MS-COCO validation set and our base-size models from the 50K-th iteration. Higher guidance scales lead to better text-aligned images at the cost of less diversity. Compared to the baseline trained only with the denoising task loss, distillation-based pretraining leads to much better trade-off curves.

### 5.2 Personalized T2I with DreamBooth

Table 5 compares the results of DreamBooth finetuning [50] with different pretrained models. BK-SDM-Small can preserve over 97% performance of the original SDM with the reduced finetuning time and number of parameters. Figure 8 depicts that our models can accurately capture the subject details and generate various scenes. Over the models pretrained with a batch size of 64, we observe the impact of KD pretraining on personalized synthesis. The baselines without KD fail to generate the subjects entirely or cannot maintain the identity details.

Table 5: Personalized generation with finetuning over different pretrained models. Our compact models can preserve subject fidelity (DINO and CLIP-I) and prompt fidelity (CLIP-T) of the original SDM with reduced finetuning (FT) time and fewer parameters.

| Pretrained Model | DINO ↑ | CLIP-I ↑ | CLIP-T ↑ | FT Time[†] | # Params |
|---|---|---|---|---|---|
| SDM-v1.4 [46, 47] | 0.728 | 0.725 | 0.263 | 881.3s | 1.04B |
| BK-SDM-Base (Ours) | 0.723 | 0.717 | 0.260 | 622.3s | 0.76B |
| BK-SDM-Small (Ours) | 0.720 | 0.705 | 0.259 | 603.6s | 0.66B |
| BK-SDM-Base, Batch Size 64 | 0.718 | 0.708 | 0.262 | 622.3s | 0.76B |
| - Without KD & Random Init. | 0.594 | 0.465 | 0.191 | 622.3s | 0.76B |
| - Without KD & Pretrained Init. | 0.716 | 0.669 | 0.258 | 622.3s | 0.76B |

[†] Per-subject finetuning time for 800 iterations on NVIDIA GeForce RTX 3090 GPU.

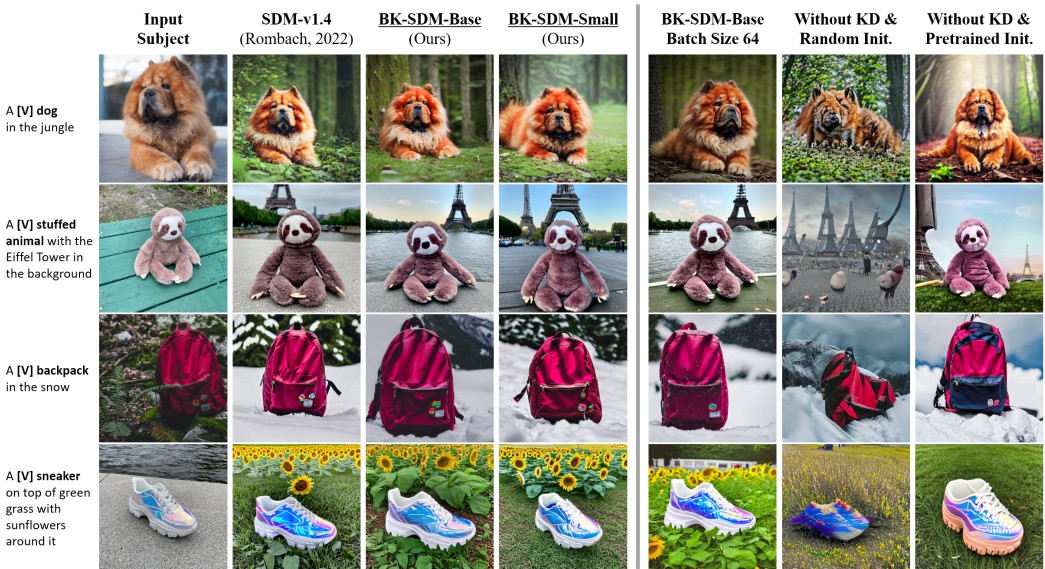

Figure 8: Visual results of personalized generation. Each subject is marked as "a [identifier] [class noun]" (e.g., "a [V] dog"). Similar to the original SDM, our compact models can synthesize the images of input subjects in different backgrounds while preserving their appearance.

## 6 Conclusion and discussion

This study uncovers the potential of architectural compression for general-purpose text-to-image synthesis with a renowned model, Stable Diffusion. Our block-removed lightweight models are effective for zero-shot generation, achieving competitive performance against large-scale baselines. Distillation is a key aspect of our method, leading to effective pretraining even under very constrained resources. Moreover, our smaller and faster pretrained models are successfully applied in personalized generation. Our work is orthogonal to previous directions for efficient diffusion models, e.g., enabling fewer sampling steps, and can be readily combined with them. We hope our study can facilitate future research on structural compression of large diffusion models.

**Limitations and future works.** Our compact models inherit the capability of the source model for high-fidelity image generation, but they have shortcomings such as inaccurate generation of full-body human appearance. While we show that distillation pretraining is powerful even with very limited resources, increasing the volume of data and analyzing its effects would be promising.

**Negative social impacts.** Because recent large generative models are capable of creating high-quality plausible content, they also involve potential risks of malicious use. To avoid causing unintended social bias, researchers should take steps to ensure the appropriateness of training data. Moreover, the release of resulting models should be accompanied by strong and reliable safeguards.

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
