# Appendix: On Architectural Compression of Text-to-Image Diffusion Models

## A  U-Net architectures of BK-SDMs

Figure 1 depicts the U-Net architectures. Compared to the 1.04B-parameter original SDM (with 0.86B-parameter U-Net), our models are smaller and lighter: 0.76B-parameter BK-SDM-Base (with 0.58B-parameter U-Net), 0.66B BK-SDM-Small (0.49B U-Net), and 0.50B BK-SDM-Tiny (0.33B U-Net). Section B introduces BK-SDM-Tiny, which is compressed further from BK-SDM-Small. Section D describes the details of block components.

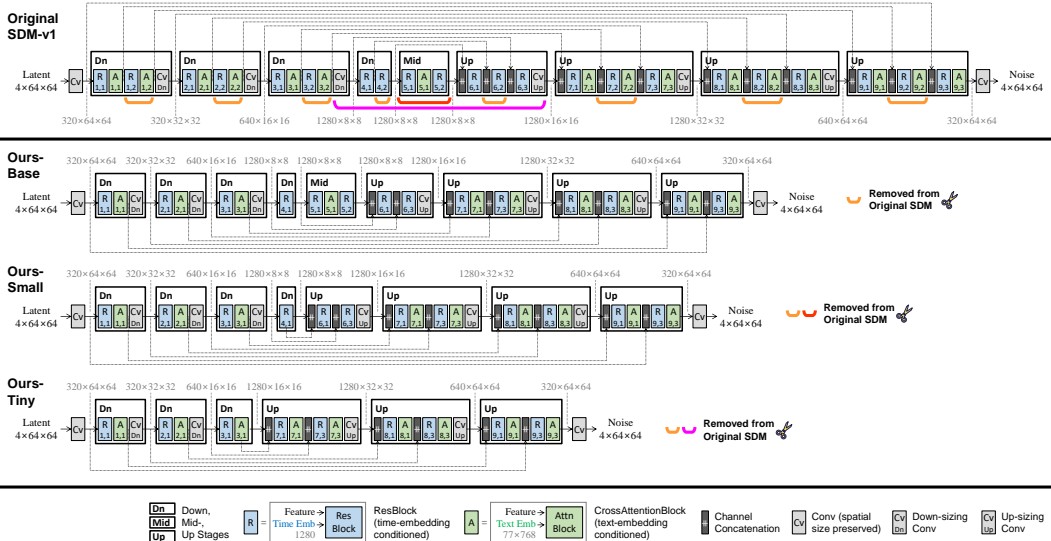

Figure 1: U-Net architectures of the original SDM-v1 and BK-SDMs.

## B  Further compression: BK-SDM-Tiny

To further improve compute efficiency, the innermost down and up stages can also be pruned (indicated with pink in Figure 1), leading to BK-SDM-Tiny. This implies that outer stages with larger spatial dimensions and their skip connections play a crucial role in the U-Net for T2I synthesis. Notably, BK-SDM-Tiny has 50% fewer parameters compared to the original SDM (see Table 1) while achieving competitive performance in zero-shot general-purpose generation (Tables 2 and 3 and Figure 2) and personalized synthesis with DreamBooth (Table 4).

Table 1: The impact of per-step compute reduction of the U-Net on the entire SDM. The number of sampling steps is indicated with the parentheses, e.g., U-Net (1) for one step. The full computation (denoted by "Whole") covers the text encoder, U-Net, and image decoder. All corresponding values are obtained on the generation of a single 512×512 image with 25 denoising steps. The latency was measured on Xeon Silver 4210R CPU 2.40GHz and NVIDIA GeForce RTX 3090 GPU.

| Model | # Params | | MACs | | | CPU Latency | | | GPU Latency | | |
|---|---|---|---|---|---|---|---|---|---|---|---|
| | U-Net | Whole | U-Net (1) | U-Net (25) | Whole | U-Net (1) | U-Net (25) | Whole | U-Net (1) | U-Net (25) | Whole |
| SDM-v1.4 [9] | 860M | 1033M | 339G | 8469G | 9716G | 5.63s | 146.28s | 153.00s | 0.049s | 1.28s | 1.41s |
| BK-SDM- Base (Ours) | 580M (-32.6%) | 752M (-27.1%) | 224G (-33.9%) | 5594G (-33.9%) | 6841G (-29.5%) | 3.84s (-31.8%) | 99.95s (-31.7%) | 106.67s (-30.3%) | 0.032s (-34.6%) | 0.83s (-35.2%) | 0.96s (-31.9%) |
| BK-SDM- Small (Ours) | 483M (-43.9%) | 655M (-36.5%) | 218G (-35.7%) | 5444G (-35.7%) | 6690G (-31.1%) | 3.45s (-38.7%) | 89.78s (-38.6%) | 96.50s (-36.9%) | 0.030s (-38.7%) | 0.77s (-39.8%) | 0.90s (-36.1%) |
| BK-SDM- Tiny (Ours) | 324M (-62.4%) | 496M (-51.9%) | 206G (-39.5%) | 5126G (-39.5%) | 6373G (-34.4%) | 3.03s (-46.2%) | 78.77s (-46.1%) | 85.49s (-44.1%) | 0.026s (-46.9%) | 0.67s (-47.7%) | 0.80s (-43.2%) |

Table 2: Zero-shot results on 30K prompts from MS-COCO validation set [4] at 256×256 resolution. Despite being trained with a smaller dataset and having fewer parameters, our compressed models achieve results on par with prior approaches for general-purpose T2I. For our models, the results with the minimum FID and the final 50K-th iteration are reported.

| Model | Type | FID ↓ | IS ↑ | # Params | Data Size |
|---|---|---|---|---|---|
| SDM-v1.4 [10] | DF | 13.05 | 36.76 | 1.04B | 600M |
| Small Stable Diffusion [6] | DF | 12.76 | 32.33 | 0.76B | 229M |
| BK-SDM-Base (Ours) @ Min FID | DF | 13.57 | 29.22 | 0.76B | 0.22M |
| BK-SDM-Base (Ours) @ Final Iter | DF | 15.76 | 33.79 | 0.76B | 0.22M |
| BK-SDM-Small (Ours) @ Min FID | DF | 15.93 | 29.61 | 0.66B | 0.22M |
| BK-SDM-Small (Ours) @ Final Iter | DF | 16.98 | 31.68 | 0.66B | 0.22M |
| BK-SDM-Tiny (Ours) @ Min FID | DF | 16.54 | 29.84 | 0.50B | 0.22M |
| BK-SDM-Tiny (Ours) @ Final Iter | DF | 17.12 | 30.09 | 0.50B | 0.22M |
| DALL·E[†][⋆] [7] | AR | 27.5 | 17.9 | 12B | 250M |
| CogView[‡][⋆] [1] | AR | 27.1 | 18.2 | 4B | 30M |
| CogView2[†][⋆] [2] | AR | 24.0 | 22.4 | 6B | 30M |
| Make-A-Scene[‡] [3] | AR | 11.84 | - | 4B | 35M |
| LAFITE[‡][♯] [12] | GAN | 26.94 | 26.02 | 0.23B | 3M |
| GALIP (CC3M)[†] [11] | GAN | 16.12 | - | 0.32B | 3M |
| GALIP (CC12M)[†] [11] | GAN | 12.54 | - | 0.32B | 12M |
| GLIDE[‡] [5] | DF | 12.24 | - | 5B | 250M |
| LDM-KL-8-G[‡][♯] [10] | DF | 12.63 | 30.29 | 1.45B | 400M |
| DALL·E-2[†] [8] | DF | 10.39 | - | 5.2B | 250M |

[†] and [‡]: FID from [11] and [10], respectively. [⋆] and [♯]: IS from [2] and [10], respectively. DF and AR: diffusion and autoregressive models. ↓ and ↑: lower and higher values are better.

Table 3: Ablation study on zero-shot MS-COCO 256×256 30K. The common settings include fewer blocks in the down and up stages and the denoising task loss. N5, N7, and N9 correspond to BK-SDM-Base, BK-SDM-Small, and BK-SDM-Tiny, respectively

| | Model | | | | | Performance | | |
|---|---|---|---|---|---|---|---|---|
| No. | Weight Initialization | Output KD | Feature KD | Batch Size | # Removed Inner Stages | FID ↓ | IS ↑ | CLIP Score ↑ |
| N1 | Random | ✗ | ✗ | 64 | ✗ | 43.80 | 13.61 | 0.1622 |
| N2 | Pretrained | ✗ | ✗ | 64 | ✗ | 20.45 | 22.68 | 0.2444 |
| N3 | Pretrained | ✓ | ✗ | 64 | ✗ | 16.48 | 27.30 | 0.2620 |
| N4 | Pretrained | ✓ | ✓ | 64 | ✗ | 14.61 | 31.44 | 0.2826 |
| N5 | Pretrained | ✓ | ✓ | 256 | ✗ | 15.76 | 33.79 | 0.2878 |
| N6 | Pretrained | ✓ | ✓ | 64 | 1 | 16.87 | 29.51 | 0.2644 |
| N7 | Pretrained | ✓ | ✓ | 256 | 1 | 16.98 | 31.68 | 0.2677 |
| N8 | Pretrained | ✓ | ✓ | 64 | 3 | 17.28 | 28.33 | 0.2607 |
| N9 | Pretrained | ✓ | ✓ | 256 | 3 | 17.12 | 30.09 | 0.2653 |
| Original SDM-v1.4 [9, 10] | | | | | | 13.05 | 36.76 | 0.2958 |

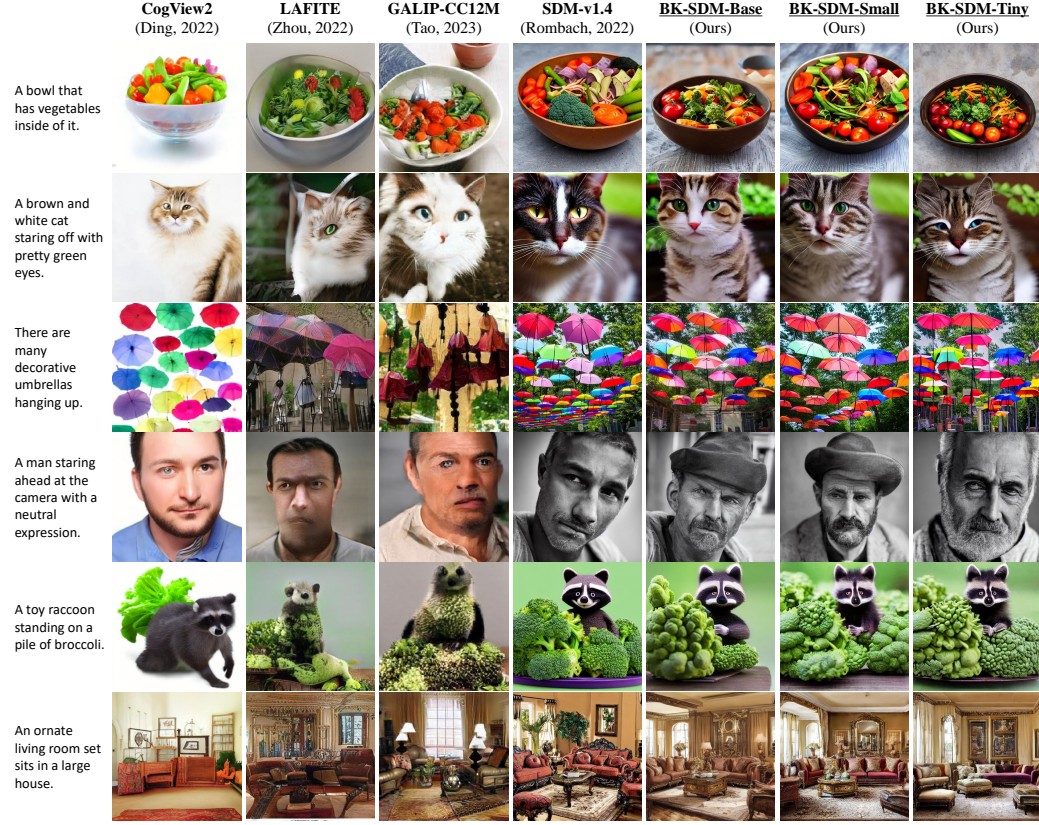

Figure 2: Visual comparison on zero-shot MS-COCO benchmark. The results of previous studies [2, 12, 11] were obtained with their official codes and released models. We do not apply any CLIP-based reranking for SDM and our models.

Table 4: Personalized generation with finetuning over different pretrained models. Our compact models can preserve subject fidelity (DINO and CLIP-I) and prompt fidelity (CLIP-T) of the original SDM with reduced finetuning (FT) cost and fewer parameters.

| Pretrained Model | DINO ↑ | CLIP-I ↑ | CLIP-T ↑ | FT Time[†] | FT Mem[‡] | # Params |
|---|---|---|---|---|---|---|
| SDM v1.4 [9, 10] | 0.728 | 0.725 | 0.263 | 881.3s | 23.0GB | 1.04B |
| BK-SDM-Base (Ours) | 0.723 | 0.717 | 0.260 | 622.3s | 18.7GB | 0.76B |
| BK-SDM-Small (Ours) | 0.720 | 0.705 | 0.259 | 603.6s | 17.2GB | 0.66B |
| BK-SDM-Tiny (Ours) | 0.715 | 0.693 | 0.261 | 559.3s | 13.1GB | 0.50B |
| BK-SDM-Base, Batch Size 64 | 0.718 | 0.708 | 0.262 | 622.3s | 18.7GB | 0.76B |
| - Without KD & Random Init. | 0.594 | 0.465 | 0.191 | 622.3s | 18.7GB | 0.76B |
| - Without KD & Pretrained Init. | 0.716 | 0.669 | 0.258 | 622.3s | 18.7GB | 0.76B |

Per-subject finetuning time[†] and GPU memory[‡] for 800 iterations with a batch size of 1 on NVIDIA GeForce RTX 3090.

## C   Impact of distillation on pretraining phase

Figure 3 shows additional results for the performance over training progress. Without KD, training compact models solely with the denoising task loss causes fluctuations or sudden drops in performance (indicated with green and cyan). Compared to the absence of KD, distillation (purple and pink) stabilizes and accelerates the training process, improving generation scores. This clearly demonstrates the benefits of providing sufficient hints for training guidance. Additionally, our small-size and tiny-size models trained with KD (yellow and red) outperform the bigger base-size models without KD (green and cyan).

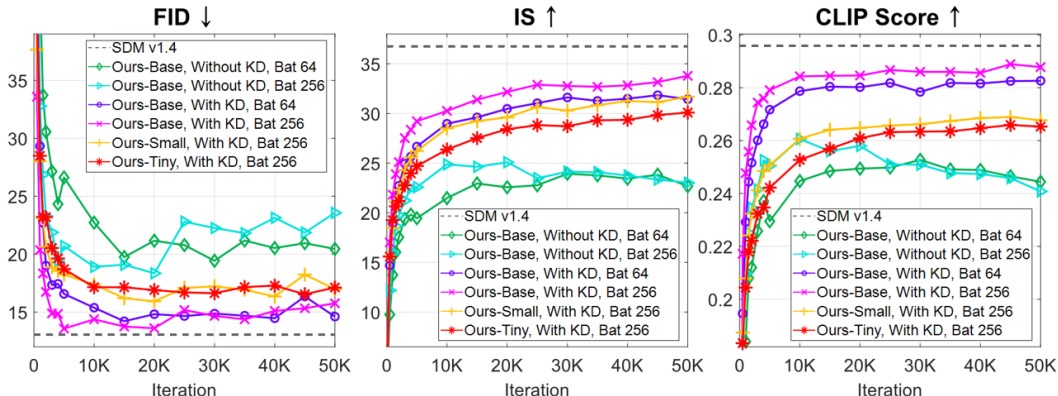

Figure 3: Results on zero-shot MS-COCO 256×256 30K over training progress. The architecture size, usage of KD, and batch size are denoted for our models.

## D   Details of block components in SDM's U-Net

Figure 4 shows the details of architectural blocks (depicted in Figure 1). Each residual block (ResBlock) contains two 3-by-3 convolutional layers and is conditioned on the time-step embedding. Each attention block (AttnBlock) contains a self-attention module, a cross-attention module, and a feed-forward network. The text embedding is merged via the cross-attention module. Within the attention block, the feature spatial dimensions H and W are flattened into a sequence length of HW. The number of channels C is considered as an embedding size, processed with 8 attention heads. The number of groups for the group normalization is set to 32. Except the down-sizing part, all the convolutional layers maintain the spatial dimensions by adjusting the stride and padding.

## E   Further implementation details

**Distillation-based Pretraining.** For augmentation, smaller edge of each image is resized to 512, and a center crop of size 512 is applied with random flip. We use a single NVIDIA A100 80G GPU for 50K-iteration pretraining with the AdamW optimizer and a constant learning rate of 5e-5. With the gradient accumulation steps of 4, the total batch size is set to either 64 or 256. With a batch size of 64 for training BK-SDM-Base, it takes about 60 hours for 50K iterations and 28GB GPU memory. With a batch size of 256, it takes about 300 hours and 53GB GPU memory. Training smaller architectures results in 5∼10% decrease in GPU memory usage.

**DreamBooth Finetuning.** For augmentation, smaller edge of each image is resized to 512, and a random crop of size 512 is applied. We use a single NVIDIA GeForce RTX 3090 GPU to finetune each personalized model for 800 iterations with the AdamW optimizer and a constant learning rate of 1e-6. We jointly finetune the text encoder as well as the U-Net part. For each subject, 200 class images are generated by the original SDM. The weight of prior preservation loss is set to 1. With a batch size of 1, the original SDM requires 23GB GPU memory for finetuning, whereas BK-SDMs require 13∼19GB memory.

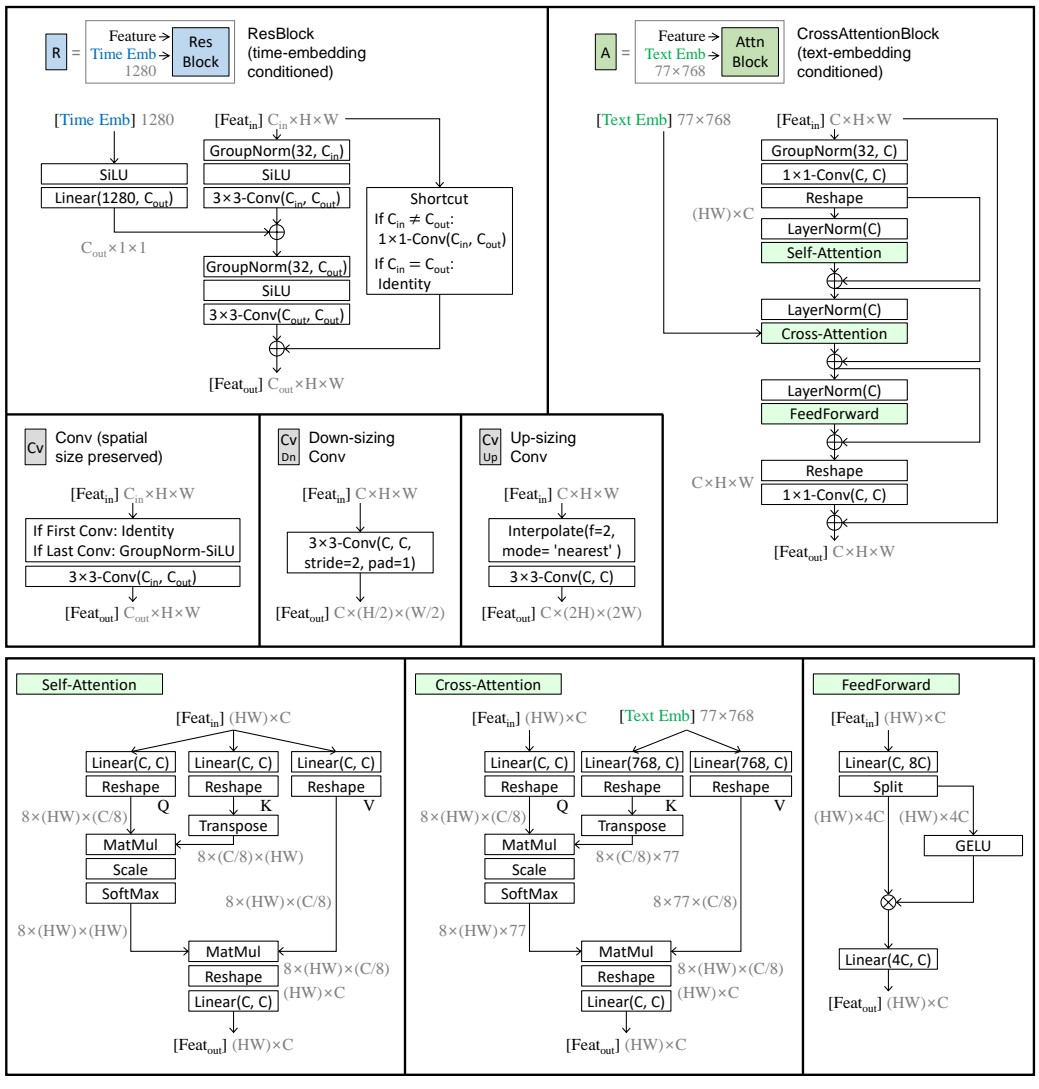

Figure 4: Block components in the U-Net of SDMs.