# OpenReview forum: "On Architectural Compression of Text-to-Image Diffusion Models"
_NeurIPS.cc/2023/Conference — Submitted to NeurIPS 2023_

### Official Review · Reviewer_wHDk · 2023-06-29

**Soundness:** 3 good
**Presentation:** 2 fair
**Contribution:** 1 poor
**Rating:** 3
**Confidence:** 4

**Summary:**

This paper proposes a comprehensive stable diffusion compression pipeline, including architecture compression and knowledge distillation. The authors apply this technique to general text-to-image generation and subject-driven image generation tasks, showing competitive performance as compared with the original stable diffusion while reducing parameter numbers and training dataset size.

**Strengths:**

1. The approach proposed in this paper is straightforward.
2. The writing of this paper is easy to understand.

**Weaknesses:**

1. The most important issue is that this paper is just a marginal improvement based on stable diffusion, so it cannot be generalized to other text-to-image synthesis frameworks, which severally narrows its application scope.
2. Reducing the SD parameter number from 1B to 0.76B or 0.66B is not so significant that this approach may not bring leaping progress in industrial applications.
3. The output and feature distillation are just classical distillation strategies that are widely adopted in other domains, e.g., image classification or GAN-based image generation. So here is a lack of innovation.

**Questions:**

1. This paper saying "this is the first study to architecturally compress large-scale diffusion models" may not be right. The paper "Scalable Diffusion Models with Transformers" discussed this issue in December 2022. Did the authors ignore this work?
Please refer to Peebles, William, and Saining Xie. "Scalable Diffusion Models with Transformers" arXiv preprint arXiv:2212.09748 (2022).
2. Is there any theoretical analysis or intuitive explanation of Sections 3.1.1 and 3.1.2, which only simply propose to reduce blocks in the down and up stages and remove the mid-stage?

---

> ### Author Rebuttal · Authors · 2023-08-07
>
> We are grateful for your valuable feedback. We kindly request that you review our responses and new experiments and consider revising your rating accordingly. If you have additional questions, please let us know.
>
> > Importance score for each block and group in SD
>
> * We appreciate your valuable comment. We newly analyze the “block-wise sensitivity” and “block-group-wise sensitivity” (see **PDF-Fig. R1**). The results are aligned with our architectural choices (e.g., removing inner blocks and first RA pairs in the down stages), clearly supporting our study. We remark that this is the first to perform a pruning sensitivity analysis for large diffusion models and hope our effort can alleviate your concern.
> * We note that block removal is not a viable action for all the blocks, because of some blocks with different channel dimensions of input and output (particularly the blocks in the later up stages). To handle this, we replace them with channel interpolation modules to mimic the removal while retaining the information.
> * Additionally, treating a ResBlock (R) and an AttentionBlock (A) as a pair and an integrated unit could be a valuable approach, given their complementary roles in processing different information types (R for time embeddings and A for text information) and the design philosophy of the original model.
>
> A portion of the block-groupwise sensitivity (please see **PDF-Fig. R1** for the block-wise sensitivity and the full groupwise analysis)
>
> | Stage | Removed/Replaced* |  FID↓ |  IS↑ | CLIP Score↑ |
> |-------|:-----------------:|:-----:|:----:|:-----------:|
> | Down  |      RA(1,1)      |  46.4 | 22.6 |    0.240    |
> | Down  |      RA(1,2)      |  35.2 | 28.2 |    0.263    |
> | Down  |      RA(2,1)*     | 226.4 |  5.1 |    0.043    |
> | Down  |      RA(2,2)      |  30.2 | 29.3 |    0.264    |
> | Down  |      RA(3,1)*     | 107.5 | 12.3 |    0.092    |
> | Down  |      RA(3,2)      |  19.0 | 32.4 |    0.286    |
> | Down  |      RR(4,1)      |  19.9 | 35.2 |    0.297    |
> | Mid   |      RAR(5,1)     |  22.1 | 31.9 |    0.264    |
> | Up    |     RRR(6,1)*     |  22.1 | 31.8 |    0.261    |
> | ...   |      ...    | ... |  ... |    ...    |
> | -     |  Original SD-v1.4 |  19.9 | 35.3 |    0.300    |
>
>
> >Regarding "Scalable Diffusion Models with Transformers"
>
> Thank you for introducing this paper; we were not aware of its existence before. However, we wish to clarify that the paper's primary focus is not on compressing pretrained diffusion models. Instead, this work suggests a transformer-based architecture to replace the U-Net and trains it without considering the original weights, with an emphasis on analyzing model scalability. Nonetheless, we will include it in our related work section, given its relevance to diffusion model architectures.
>
> > Further compression and importance of this study
>
> * Please find another architectural variation (**BK-SDM-Tiny** with further inner-stage removal) presented in Supplementary Section B, which showcases an impressive compression rate (**51% fewer parameters**, 43% latency improvement).
> * We emphasize that the substantial compute resources for training SD (e.g., **256 A100 GPUs and 600M LAION pairs** for SD-v1.4) are unfeasible for most researchers. Our work demonstrates that it is possible to compress SD with very limited resources (**1 A100 GPU and 0.22M pairs**) by benefiting from transferred knowledge.
> * We have already demonstrated the successful utilization of our models in personalized generation (Sections B and 5.2), an application of great public interest. This achievement was attained not only with reduced parameters and latency but also with lower fine-tuning costs.
>
> > Range of Applications
>
> * Thank you for sharing your view. However, the Stable Diffusion family is one of the rarely open-sourced foundation models for text-to-image synthesis, which has brought about a paradigm shift in both research and industrial communities. To the best of our knowledge, other multibillion diffusion models with impressive performance are not publicly available.
> * As described in Section 4.2, our work can be applicable to other SD versions in v1.1–v1.5 with the same architecture and to SDM-v2 with a similarly designed architecture (consisting of RA pairs).
> * Furthermore, we also showcase the versatility of our approach in general image generation (please refer to **PDF-Fig. R4**). The architectural design of SD's U-Net is broadly used (as mentioned in the LDM paper, "general-purpose conditioning mechanism based on cross-attention"), indicating the extensive applicability of our method.
>
> > Comparison to previous compression studies
> * To the best of our knowledge, prior compression studies on CNNs and Transformers have not dealt with such large models: e.g., ResNet152 (with 60.2M parameters) and ViT_L_16 (306M params) are much smaller than SD-v1.4 (1033M params). Moreover, U-Net architectures are arguably more complex due to the necessity of considering skip connections across the network, making the structural removal inside them not straightforward.
> * In the age of foundation models, we highlight the increasing significance of technical/empirical contributions for large models. For example, in the Google’s Imagen paper [NeurIPS’22], their key discoveries regarding the model size and training corpus for the text encoder are important in developing future T2I models. We believe our study offers value in advancing small general-purpose models, by revealing the astonishing effectiveness of architecture compression.
> * Moreover, previous studies using distillation for fewer steps of diffusion models are highly valuable but cannot reduce the model size (because of distilling over identically structured models), which our architectural compression can address. We emphasize that the potential of Feature KD (a crucial element for cost-effective training) remains unexplored in earlier studies of diffusion models.

---

> > ### Author Response · Authors · 2023-08-21
> >
> > Dear Reviewer wHDk,
> >
> > Thank you for your time and effort in reviewing our work.
> >
> > The author-reviewer discussion ends soon, and we would like to confirm whether we've addressed your concerns. Could you please take a look at our rebuttal?
> >
> > Thank you,
> >
> > Authors

---

> > ### Comment · Reviewer_wHDk · 2023-08-21
> >
> > Dear authors,
> >
> > Sorry for my late response. Actually, we have reimplemented your paper under the BK-SDM-Small setup and trained on a small dataset (600K image-text pairs). We finally achieve 43% model size compression and 4 FID degradation, which matches your paper claim. So we absolutely believe in and appreciate your paper value.
> >
> > I have also carefully read all the reviewers' comments and the heated discussion. I still think the main contribution of the paper—architecture compression—needs more in-depth analysis, especially for the scalability of the re-designed UNet structure. For example, can we compress it to 10% of the original Unet size to achieve a fast running speed even though it is of low generation quality? But the image quality is enough to provide a judgment for users to decide whether to use the original model to regenerate images with the same random seeds.
> >
> > I have to keep my score. But I wish to see a revised version of this paper with more in-depth analysis and more compression choices.

---

> > > ### Author Response · Authors · 2023-08-21
> > >
> > > We sincerely appreciate your response despite your busy schedule. We are delighted to know that you have successfully re-implemented our paper and recognized its value. Thank you.
> > >
> > > Given the below factors, we humbly express our concern about the severity of the assigned rating, 'Rating: 3: Reject'. In light of the contributions our work brings to the field, we respectfully request you to reconsider the rating.
> > >
> > > Thank you very much for your dedication to this process.
> > >
> > > Best,
> > >
> > > Authors
> > >
> > > ---
> > >
> > > Regarding the architectural variations, we would like to emphasize the presentation of **BK-SDM-Tiny** (with 51% fewer parameters and 43% latency improvement) that ensures acceptable generation quality. Moreover, we have introduced **several architecture variants by removing less critical blocks** from the sensitivity analysis (PDF-Fig. R1(a)). It would be appreciated if you could take a look at our response to Reviewer Gqvg; please understand that we do not attach the details to avoid excessive length of this response.
> > >
> > > ---
> > >
> > > > Can we compress it to 10% of the original Unet size to achieve a fast running speed even though it is of low generation quality?
> > >
> > > It looks intriguing, and we will elaborate on the effect of more excessive compression rates. However, prioritizing generation quality over inference speed may be more important, because this is the main reason people love the power of large generative models despite their considerable compute.
> > >
> > > Furthermore, to the best of our knowledge, numerous previous studies on compressing generative models (such as GAN compression) have not conducted such analyses, and we respectfully express our opinion that this particular factor should not be the important criterion for evaluating our work.
> > >
> > > ---
> > > We would like to attach `Reviewer Gqvg's global comments`:
> > >
> > > - In my personal view, technical novelty in the sense of having to produce some arbitrary (and often complex for the sake of novelty) algorithm should not be a requirement for a paper to be accepted.
> > > - I see this paper in the same way. The authors provide an incremental, but still useful measurement study (especially with the additional block-wise sensitivity analysis they provide) that will guide practitioners who work in enabling text-to-image generation on more resource constrained settings.
> > > - While incremental, this is an impactful analysis because there is an immediate real-world need where even a 50% decrease in parameters can significantly increase the number of legacy GPUs that people can run text-to-image generation on. This is also something that is currently performed in ad-hoc fashion and hidden in the depths of Discord servers of text-to-image gurus without real evidence or structured analysis; making this analysis instead available to the wider community and in the open is a definite need.
> > > - This paper will also be sure to generated follow-on works that will use the results of the analysis they provide, and either 1. come up with some theoretical explanations that lead in the sensitivities they see or 2. an algorithm that can provide a general pruning mechanism. I don't think this paper necessarily needs to provide all of these things at once.
> > >
> > > ---
> > >
> > > We strongly believe that empirical contributions become even more crucial in the era of large foundational models. Our study, which *proposes appropriate compressed architectures for Stable Diffusion, uncovers the potential of feature KD, presents valuable analyses (#), and makes experiments feasible for individual researchers*, can greatly benefit the community.
> > >
> > > - (#) Our work includes the MS-COCO benchmark, ablation study, performance change over training progress, effect of classifier-free guidance scales, effect of different numbers of sampling steps, effect of training data volume, block-level sensitivity analyses, and application to unconditional image generation. We would greatly appreciate it if you could take these aspects into consideration.

---

### Official Review · Reviewer_Gqvg · 2023-07-04

**Soundness:** 4 excellent
**Presentation:** 4 excellent
**Contribution:** 3 good
**Rating:** 7
**Confidence:** 3

**Summary:**

This paper studies model compression for stable diffusion models (1.4 specifically). Specifically, they study 1. removing entire layers from the diffusion U-Net, 2. fine-tuning from a small subset of LAION to recover the loss, and 3. knowledge distillation from intermediate feature layers. They show that they can get a significant (~30-40%) reduction in both parameter count and inference latency at a modest output quality degradation, at a modest training cost (60-300 hours on a single GPU) that individuals can have access to.

**Strengths:**

The paper works on a timely topic, especially given that Stable Diffusion has widespread use in the open source community, and reducing the inference cost will not only have a broad impact on usability for these models, but also the environment. The paper is easy to understand, and seems to be first to evaluate structured pruning (broadly) on Stable Diffusion to the best of my knowledge. The analysis includes several different metrics and ablation studies that are useful to the community.

**Weaknesses:**

I found it hard to intuitively understand why the layers were specifically chosen to remove. One of the interesting results the authors show in this paper is that removing the mid-layer without any additional fine tuning already results in decent quality (competitive with some of the smaller models that are fine-tuned). Then, a natural question that might come up then is if we were to remove every layer and run an analysis of how much cost (in quality loss) the removal of each layer ends up with, would we then be able to confirm that the layers that were removed are in fact the best layers to remove in terms of their quality cost? This would be much cheaper of an experiment to run than fine-tuning the entire removed architecture, and would be very helpful to know if the "unsupervised" structured pruning is a good indicator for the end performance after fine-tuning.

It would at least be interesting if there could be an analysis of how much the fine-tuning (and KD) adds (in terms of quality) to an 'unsupervised' baseline, whether that means running an ablation specifically with the mid results removed, or adding an 'unsupervised' baseline with all layers removed (the orange and red in Figure 3).

This is very much concurrent work (and does not affect my score), but it could be worth adding https://arxiv.org/abs/2305.10924 to the citations.

**Questions:**

1. How costly is the addition of KD in terms of training times? The supplementary mentions training times but not with respect to some of these design decisions (as far as I could find)



**Limitations:**

The limitations are adequately addressed. It might also be interesting if the authors can comment whether model compression itself could have risks in society, for example if people prefer worse but slightly faster models over better models, then that could potentially exacerbate the bias that these models have (and of the models people use in practice).

---

> ### Author Rebuttal · Authors · 2023-08-07
>
> We sincerely appreciate your positive and valuable feedback. The inclusion of the experimental results you advised has significantly improved our paper. Feel free to reach out to us if you have any additional questions.
>
> > Importance score for each block and group in SD
>
> * We appreciate your valuable comment. We newly analyze the “block-wise sensitivity” and “block-groupwise sensitivity” (see **PDF-Fig. R1** and the below table). The results are aligned with our architectural choices (e.g., removing inner blocks and first RA pairs in the down stages), supporting our study. We remark that this is the first to perform a pruning sensitivity analysis for large diffusion models.
> * We note that block removal is not a viable action for all the blocks, because of some blocks with different channel dimensions of input and output (particularly the blocks in the later up stages). To handle this, we replace them with channel interpolation modules to mimic the removal while retaining the information.
> * Additionally, treating a ResBlock (R) and an AttentionBlock (A) as a pair and an integrated unit could be a valuable approach, given their complementary roles in processing different information types (R for time embeddings and A for text information) and the design philosophy of the original model.
>
> A portion of the block-groupwise sensitivity (please see **PDF-Fig. R1** for the block-wise sensitivity and the full groupwise analysis)
>
> | Stage | Removed/Replaced* |  FID↓ |  IS↑ | CLIP Score↑ |
> |-------|:-----------------:|:-----:|:----:|:-----------:|
> | Down  |      RA(1,1)      |  46.4 | 22.6 |    0.240    |
> | Down  |      RA(1,2)      |  35.2 | 28.2 |    0.263    |
> | Down  |      RA(2,1)*     | 226.4 |  5.1 |    0.043    |
> | Down  |      RA(2,2)      |  30.2 | 29.3 |    0.264    |
> | Down  |      RA(3,1)*     | 107.5 | 12.3 |    0.092    |
> | Down  |      RA(3,2)      |  19.0 | 32.4 |    0.286    |
> | Down  |      RR(4,1)      |  19.9 | 35.2 |    0.297    |
> | Mid   |      RAR(5,1)     |  22.1 | 31.9 |    0.264    |
> | Up    |     RRR(6,1)*     |  22.1 | 31.8 |    0.261    |
> | ...   |      ...    | ... |  ... |    ...    |
> | -     |  Original SD-v1.4 |  19.9 | 35.3 |    0.300    |
>
> * We further analyze different architectures by removing less important blocks in **PDF-Fig. R1(a)**. The CLIP scores are sorted and used to remove the top 14 blocks (with CLIP score > 0.29), 21 blocks (CLIP score > 0.265), 24 blocks (CLIP score > 0.245 and FID < 25), and 26 blocks (CLIP score > 0.20 and FID < 100). The post-pruning results are as follows, while training these architectures remains an interesting future direction. While it remains uncertain whether extensively eliminating many attention blocks (which handle text embeddings) can result in good text-aligned images, these could serve as valuable baselines for our models. Thank you once again for your comment.
>
> Post-pruning results (without training) on MS-COCO 512×512 5K
>
> | Architecture     | # Params, Whole (U-Net) | FID↓  | IS↑ | CLIP Score↑ |
> |------------------|-------------------------|-------|-----|-------------|
> | Remove 14 blocks | 781M (609M)             | 135.1 | 6.9 | 0.1226      |
> | Remove 21 blocks | 604M (431M)             | 274.6 | 2.0 | 0.0115      |
> | Remove 24 blocks | 456M (284M)             | 274.9 | 2.0 | 0.0120      |
> | Remove 26 blocks | 452M (279M)             | 348.6 | 1.2 | 0.0797      |
> | BK-SDM-Base      | 752M (580M)             | 537.9 | 2.1 | 0.0012      |
> | BK-SDM-Small     | 655M (483M)             | 539.4 | 2.1 | 0.0009      |
> | BK-SDM-Tiny      | 496M (324M)             | 537.4 | 2.2 | 0.0007      |
>
>
> > Adding a related study
>
> Thank you for introducing this relevant work, Diff-Pruning. We will add this work to the related work.
>
> > Reporting training times
>
> * Thanks for your question. Pretraining BK-SDM-Base with a batch size of 256 for 50K iterations takes 272 hours without KD and 301 hours with KD.
> * Furthermore, under the same setup with KD, pretraining BK-SDM-Small and BK-SDM-Tiny takes 294 and 288 hours, respectively. While the reduction in training time might not be significant, the noteworthy gains lie in our model's inference compute efficiency and superior performance.
>
> > Regarding a further ablation study
>
> * Thank you for your detailed comment, but we believe some ablation experiments you mentioned could be found in Supplementary Table 3 (also attached below) and Supplementary Figure 3. We present a thorough analysis using BK-SDM-Base with different batch sizes, to showcase the merits of KD in comparison to the baseline using only the task denoising loss (without KD). The effect of removing inner stages could be examined by comparing BK-SDM-Base and BK-SDM-{Small, Tiny}.
> * Introducing the baseline without KD for BK-SDM-Small would be valuable, but we were unable to complete the experiment within the rebuttal time limit because of handling multiple experiments. We hope you can kindly understand this point.
>
> Supplementary Table 3 (Ablation study on zero-shot MS-COCO 256×256 30K)
> | BK-SDM  (# Params) | KD | Batch Size | # Removed Inner Stages |  FID↓ |  IS ↑ | CLIP Score↑ |
> |:------------------:|:--:|:----------:|:----------------------:|:-----:|:-----:|:-----------:|
> |    Base (0.76B)    |  X |     64     |            X           | 20.45 | 22.68 |    0.2444   |
> |    Base (0.76B)    |  O |     64     |            X           | 14.61 | 31.44 |    0.2826   |
> |    Base (0.76B)    |  X |     256    |            X           | 23.57 | 23.02 |    0.2408   |
> |    Base (0.76B)    |  O |     256    |            X           | 15.76 | 33.79 |    0.2878   |
> |    Small (0.66B)   |  O |     64     |            1           | 16.87 | 29.51 |    0.2644   |
> |    Small (0.66B)   |  O |     256    |            1           | 16.98 | 31.68 |    0.2677   |
> |    Tiny (0.50B)    |  O |     64     |            3           | 17.28 | 28.33 |    0.2607   |
> |    Tiny (0.50B)    |  O |     256    |            3           | 17.12 | 30.09 |    0.2653   |

---

### Official Review · Reviewer_ZGLR · 2023-07-06

**Soundness:** 2 fair
**Presentation:** 2 fair
**Contribution:** 2 fair
**Rating:** 3
**Confidence:** 5

**Summary:**

The paper presents an technique to compress the network structure of a diffusion model. Specifically, the method combines a static block pruning strategy and a knowledge distillation retraining to learn compact diffusion model. Experiments on text to image generation and customization shows the effectiveness of the approach.

**Strengths:**

1. The paper shows that conventional compression techniques leveraged in CNNs/GANs can also be used for diffusion model.
2. Some experiments are conducted to show the effectiveness of the method.

**Weaknesses:**

1. The paper is not novel at all. Pruning and distillation has long been used for vision network compression since the era of CNNs and later GANs. Nothing surprising comes out from this paper, where the author just simply combine two techniques and makes it work. Block pruning and output + feature distillation is developed far long ago and naively combines them show little academic value.
2. The paper lacks insight significantly. There's no theoretical proof nor ablation study to judge the design choice. For example, why shall we remove the second pairs of R-A instead of the first pair of A-R in down blocks? Why shall we retain the third R-A pair in the up blocks? Why can't we do selective removal where some blocks only remove R and some blocks only remove A? The pruning design choice is totally ad-hoc where nothing systematic and insightful comes out. Due to this, the method seems to only be applicable on SD-v1.4. I see no way that how this method can be applied to SD-v2.1 / SDXL or any other new SDs where we simply add one/two more modules of R/A into a block.
3. The use of distillation-based pre-training sounds weird. Training/fine-tuning with distillation to recover the performance is a more appropriate wording.
4. In Sec 4.1, the paper says that the BK-SDM is only fine-tuned on 0.22M image-text pairs. Why only fine-tuned on such smaller scale of dataset? How about fine-tuning on 2.2M or 22M? Would the quality get improved? How about 22k or even fewer? Will the quality stay the same? No judgement is presented here as well.
5. In Figure 5, the paper should compare more diffusion method instead of showing the first 3 column of GAN-based approach. Since the speed of the network falls into the scale of diffusion model, it does not make much sense to compare GANs.

**Questions:**

Please see the Weakness section.

**Limitations:**

Yes, the author addressed both.

---

> ### Author Rebuttal · Authors · 2023-08-06
>
> We deeply appreciate your valuable feedback. The paper has been significantly strengthened by incorporating the new results from the experiments you recommended. We kindly request you to consider raising your score after reviewing our responses. If you have any further questions, please let us know.
>
> > Importance score for each block and group in SD
>
> * We appreciate your valuable comment. We newly analyze the “block-wise sensitivity” and “block-groupwise sensitivity” (see **PDF-Fig. R1** and the below table). The results are aligned with our architectural choices (e.g., removing inner blocks and first RA pairs in the down stages), clearly supporting our study. We remark that this is the first to perform a pruning sensitivity analysis for large diffusion models.
> * We note that block removal is not a viable action for all the blocks, because of their different channel dimensions of input and output (particularly for the residual blocks in the later up stages). To handle this, we replace them with channel interpolation modules to mimic the removal while retaining the information.
> * Additionally, treating a ResBlock (R) and an AttentionBlock (A) as a pair and an integrated unit could be a valuable approach, given their complementary roles in processing different information types (R for time embeddings and A for text information) and the design philosophy of the original model.
>
> A portion of the block-groupwise sensitivity (please see **PDF-Fig. R1** for the block-wise sensitivity and the full groupwise analysis)
>
> | Stage | Removed/Replaced* |  FID↓ |  IS↑ | CLIP Score↑ |
> |-------|:-----------------:|:-----:|:----:|:-----------:|
> | Down  |      RA(1,1)      |  46.4 | 22.6 |    0.240    |
> | Down  |      RA(1,2)      |  35.2 | 28.2 |    0.263    |
> | Down  |      RA(2,1)*     | 226.4 |  5.1 |    0.043    |
> | Down  |      RA(2,2)      |  30.2 | 29.3 |    0.264    |
> | Down  |      RA(3,1)*     | 107.5 | 12.3 |    0.092    |
> | Down  |      RA(3,2)      |  19.0 | 32.4 |    0.286    |
> | Down  |      RR(4,1)      |  19.9 | 35.2 |    0.297    |
> | Mid   |      RAR(5,1)     |  22.1 | 31.9 |    0.264    |
> | Up    |     RRR(6,1)*     |  22.1 | 31.8 |    0.261    |
> | Up    |      RA(7,1)*     | 103.0 | 10.9 |    0.089    |
> | Up    |      RA(7,2)*     | 135.6 |  8.1 |    0.041    |
> | Up    |      RA(7,3)*     | 132.9 |  6.9 |    0.041    |
> | ...   |      ...    | ... |  ... |    ...    |
> | -     |  Original SD-v1.4 |  19.9 | 35.3 |    0.300    |
>
>
> > Varying the training data volume
>
> Thank you for your constructive comment. We newly vary the data size from 212K pairs to {2256K, 1128K, 100K, 50K, 11K} for training BK-SDM-Small and report their impact in **PDF-Fig R2** and the below table. Increasing the number of training pairs improves the IS and CLIP scores over training progress and the visual results. We hope our effort can alleviate your concern.
>
> Evaluation on zero-shot MS-COCO 256×256 30K (the results at the final 50K-th iteration)
>
> |  Data Size  |   11K   |   50K   |   100K  |   212K  |  1128K  |  2256K  |
> |:-----------:|:-------:|:-------:|:-------:|:-------:|:-------:|:-------:|
> |    FID↓     |  18.73  |  19.03  |  17.00  |  16.98  |  16.74  |  17.05  |
> |     IS↑     |  28.75  |  31.31  |  31.17  |  31.68  |  32.84  |  33.10  |
> | CLIP Score↑ | 0.2600  | 0.2661  | 0.2697  | 0.2677  | 0.2735  | 0.2734  |
>
>
> > Clarifying the comparison in Figure 5 and adding new results
>
> * We would like to clarify that CogView2 (used in the 1st column of Figure 5) is an autoregressive model. Furthermore, to the best of our knowledge, most multibillion diffusion models with outstanding performance are not publicly available.
> * In **PDF-Fig. R5**, we add the comparison to the filtered-data GLIDE model [OpenAI, 2022], the only public diffusion model we can readily access. Our BK-SDM-Small with 0.66B parameters shows better results than the filtered GLIDE with 0.79B parameters, particularly in terms of text-image alignment.
>
> > Terminology regarding distillation-based pretraining
>
> Thanks for your careful comment. The terminology is based on DistillBERT (a famous small general-purpose language model) [NeurIPS Workshop, 2019] and the related studies in Section 2. However, we can get your point and will further clarify the selection of our terminology in the updated manuscript.
>
> > The importance of this study
>
> * To the best of our knowledge, previous compression studies on CNNs and Transformers have not dealt with such large models: e.g., ResNet152 (with 60.2M parameters) and ViT_L_16 (306M params) are much smaller than SD-v1.4 (1033M params). Moreover, prior works using distillation for fewer steps towards efficient diffusion are highly valuable but cannot reduce the model size (because of distilling over identically structured models), which our architectural compression can address.
> * In the age of foundation models, we highlight the increasing significance of technical/empirical contributions for large models. The potential of Feature KD (a crucial element for cost-effective training) remains unexplored in earlier studies of diffusion models. We believe our study offers value in advancing small general-purpose models, by revealing the astonishing effectiveness of architecture compression.
> * Nevertheless, we clarify that block removal from the huge, complex U-Net is not straightforward in terms of considering their weight connectivity. Through this rebuttal, we further justify our architectures via block-level sensitivity analyses.
>
> > Range of Applications
>
> * As described in Section 4.2, our approach can be applicable to SDM v1.1–v1.5 with the same structure and to SDM-v2 with a similarly designed architecture (consisting of RA pairs). SDXL is not the model released at the time of submission.
> * We also showcase the versatility of our approach in general image generation (**PDF-Fig. R4**). The U-Net architecture of many Latent Diffusion models resembles that of SD, indicating the extensive applicability of our method.

---

> > ### Comment · Reviewer_ZGLR · 2023-08-16
> > **Response**
> >
> > Many thanks for the author's rebuttal. The provided block analysis makes sense but still the paper needs to propose a more systematic approach to decide blocks to prune and/or pruning ratios. There's also too much modification needed and the paper in the current shape is not ready for submission. I'll therefore maintain my rating.

---

> > > ### Author Response · Authors · 2023-08-16
> > > **Thank you,**
> > >
> > > Dear Reviewer ZGLR,
> > >
> > > Thank you for taking the time to consider our rebuttal amidst your busy schedule and for sharing your valuable insights. While we hold your opinion in high regard, we kindly ask for your understanding regarding the extensive efforts we have undertaken (e.g., block-level sensitivity analysis, varying data volumes, and introducing a new baseline) to address the concerns you raised.
> > >
> > > We would like to emphasize the pioneering perspective of our work and the significance of the experiments and analyses, which would benefit the research community. Given these factors, we humbly express our concern about the severity of the assigned rating, 'Rating: 3: Reject'. In light of the contributions our work brings to the field, we respectfully request you to reconsider the rating.
> > >
> > > Regarding the incorporation of the newly conducted experiments into the manuscript, we believe that minimal modifications would be necessary, as these additions align with our current architectural choices. We once again extend our gratitude for your fruitful review and your dedication to this process.

---

### Official Review · Reviewer_M23o · 2023-07-07

**Soundness:** 4 excellent
**Presentation:** 3 good
**Contribution:** 2 fair
**Rating:** 4
**Confidence:** 4

**Summary:**

This submission proposes a light-weighted network for stable-diffusion-based text-to-image generation.
The proposed network is named BK-SDM, which became light-weighted thanks to removing unnecessary layers
and knowledge distillation.
An interesting finding is that removal of the entire mid-stage (the most low-resolution part of the u-net) does not seriously harm
generation quality.
As a result, 30% computational cost reduction was achieved while retaining 97% scores of generated image quality.


**Strengths:**

- Light-weight diffusion models are, despite their importance,  not explored. I appreciate pioneering this direction.
- Training using a small subset (0.22M image-text pairs from LAION) is a nice point that is easier to reproduce, while keeping the generation quality.

**Weaknesses:**

-  Methodological novelty is limited because layer removing and knowledge distillation are well-used technics, while they were not done in diffusion. I did not find special challenges in applying them to diffusion models.

- Trades-off between speed and quality are not aggressively investigated; the only presented model is the  30% faster / 97% worse version. For example, what happens when the model is 90% faster? Is it possible to achieve any speed-up with 100% quality? Perhaps too aggressive speed-up results in the uselessly worse generation quality, but trying and clarifying it would be a useful contribution for a paper.

**Questions:**

"0.22M image-text pairs from LAION-Aesthetics157 V2 6.5+": is this a randomly sampled version of the dataset? Or any systematic image reduction (for example, smallifying the caption vocabulary size) was done?

**Limitations:**

Limitations and potential social impacts are properly discussed.

---

> ### Author Rebuttal · Authors · 2023-08-06
>
> We appreciate your feedback. We hope our responses convince you about the significance of our approach. We kindly request that you review our responses and new experiments and consider increasing your rating accordingly. If there are any additional questions or concerns, please feel free to share them with us.
>
> >The importance of this study
>
> Thank you for sharing your opinion. We would like to emphasize the below points and hope to alleviate your concern.
> * (i) In the age of foundation models, we highlight the increasing significance of technical/empirical contributions for large models. For example, in the Google’s Imagen paper [NeurIPS’22], their key discoveries regarding the model size and training corpus for the text encoder are important in developing future T2I models. We believe our study offers value in advancing small general-purpose models, by revealing the astonishing effectiveness of architecture compression.
> * (ii) Moreover, the potential of Feature KD (a crucial element for cost-effective training) remains unexplored in earlier studies of diffusion models. Thanks to your comment, we rethink about the reason why Feature KD is particularly useful for diffusion models: the problem of estimating noises given (latent) inputs and time steps is extremely challenging, and guiding the network with sufficient hints in multiple feature levels via KD is indeed essential. This is also well demonstrated in our ablation study in Table 4.
> * (iii) Nevertheless, we clarify that block removal from the huge, complex U-Net is not technically easy and not straightforward in terms of considering their weight connectivity. Through this rebuttal, we further justify our architectural choices via **block-level sensitivity analyses** (please refer to **PDF-Fig. R1**), which are the first analyses about the importance of each block and group in Stable Diffusion.
>
> >Further trades-off between speed and quality
>
> Please find another architectural variation (**BK-SDM-Tiny** with further inner-stage removal) presented in Supplementary Section B, which showcases an impressive compression rate (**51% fewer parameters**, 43% latency improvement).
> * We believe we have extensively studied the effect of different compression rates by introducing BK-SDM-{Base, Small, Tiny} with {27%, 36%, 51%} reduced model size in Supplementary Tables 1, 2 and 3. We hope this detailed report could address your concern.
>
> Nevertheless, based on your insightful suggestion, we further analyze different architectures by removing less important blocks in **PDF-Fig. R1(a)**. Specifically, the CLIP scores are sorted and used to remove the top 14 blocks (with CLIP score > 0.29), 21 blocks (CLIP score > 0.265), 24 blocks (CLIP score > 0.245 and FID < 25), and 26 blocks (CLIP score > 0.20 and FID < 100). We observe that each of the top 21 blocks has the same input and output dimensions, and they predominantly consist of attention blocks. The post-pruning results are as follows, while the exploration of training these architectures remains an interesting future direction. While it remains uncertain whether extensively eliminating the majority of attention blocks (which handle text embeddings) can result in good text-aligned images, these could serve as valuable baselines for our models. Thank you once again for your comment.
>
> Post-pruning results (without training) on MS-COCO 512×512 5K
>
> | Architecture     | # Params, Whole (U-Net) | FID↓  | IS↑ | CLIP Score↑ |
> |------------------|-------------------------|-------|-----|-------------|
> | Remove 14 blocks | 781M (609M)             | 135.1 | 6.9 | 0.1226      |
> | Remove 21 blocks | 604M (431M)             | 274.6 | 2.0 | 0.0115      |
> | Remove 24 blocks | 456M (284M)             | 274.9 | 2.0 | 0.0120      |
> | Remove 26 blocks | 452M (279M)             | 348.6 | 1.2 | 0.0797      |
> | BK-SDM-Base      | 752M (580M)             | 537.9 | 2.1 | 0.0012      |
> | BK-SDM-Small     | 655M (483M)             | 539.4 | 2.1 | 0.0009      |
> | BK-SDM-Tiny      | 496M (324M)             | 537.4 | 2.2 | 0.0007      |
>
>
> >Construction of the dataset
>
> Thanks for your question. The image-text pairs were **randomly** sampled from LAION-Aesthetics V2 6.5+ without any curation. Moreover, we present additional results about the effect of training data size (where the pairs were also randomly sampled), and please see **PDF-Fig. R2**.

---

> > ### Comment · Reviewer_M23o · 2023-08-17
> >
> > I appreciate the Authors' response and detailed additional results.
> >
> > After looking at the additional results, I still have feelings of lacking a strong conclusion.
> > I checked Fig 2. in Supplementary Material,  but I could not grasp key differences in the generation results by Base/Small/Tiny visually.
> > It is impressive that excessive random subsampling from LAION-Aesthetics worked,
> > but I still suspect it may bring missed shortcomings; for example, in prompting with low-frequency words.
> > It would be better to have in-depth analyses on this point.
> >
> > Overall, I think another round of review is needed to inspect the reorganized manuscript with the additional results, and it would be suitable for the next venue.
> > I am going to keep my initial rating.

---

> > > ### Author Response · Authors · 2023-08-17
> > > **Thank you,**
> > >
> > > Dear Reviewer M23o,
> > >
> > > We sincerely appreciate your time in reviewing our response and the additional results. While Main Table 2 and Supplementary Table 2 have showed the quantitative results on the famous MS-COCO benchmark, we acknowledge your suggestion regarding the visual results.
> > > - As described in ‘Limitations of Sect. 6. Conclusion’, our compact models have shortcomings such as inaccurate synthesis of full-body appearance and fonts. While these limitations are also present in the original source model, they are magnified at higher compression rates due to the model's limited capacity and reduced training data. We will elaborate further on these aspects in our revised manuscript.
> > >
> > > However, we would like to clarify that we believe we have thoroughly analyzed **the behavior of our compressed models**, much like the previous studies in Table 2 (i.e., DALLE-E [ICML’21], CogView [NeurIPS’21], Cogview2 [NeurIPS’22], Make-A-Scene [ECCV’22], LAFITE [CVPR’22], GALIP [CVPR’23], GLIDE [ICML’22], DALLE-2 [arXiv’22])
> > > - Our work includes not only the MS-COCO benchmark results but also **the ablation study, the performance over training progress, the effect of classifier-free guidance scales, the effect of different numbers of sampling steps, and the block-level sensitivity analysis**. We would greatly appreciate it if you could take these aspects into consideration.
> > >
> > > We also wish to emphasize that the renowned MS-COCO benchmark covers a wide range of topics, spanning from human actions to everyday objects. The analysis involving prompts containing low-frequency words is intriguing; however, as far as we are aware, none of the prior works have conducted such an analysis, and we respectfully express our opinion that this particular factor should not be the important criterion for evaluating our work.
> > >
> > > Nevertheless, we gratefully acknowledge that your viewpoints and feedback enhance the depth of our research. We hold your dedication in high regard and express our genuine gratitude. We will put in the effort to incorporate the needed revisions into our manuscript. Thank you.
> > >
> > > Best regards,
> > >
> > > Authors

---

### Official Review · Reviewer_WEDt · 2023-07-07

**Soundness:** 3 good
**Presentation:** 3 good
**Contribution:** 2 fair
**Rating:** 5
**Confidence:** 4

**Summary:**

This work explores traditional model compression for diffusion models, aiming to mitigate their considerable computational overhead. By employing block removal and knowledge distillation, the authors are able to reduce parameters by over 30% with only minimal performance loss with the 0.22M LAION dataset. The paper further showcases its applicability for personalized generation tasks.

**Strengths:**

1. The proposed hand-crafted compression strategy, that leverages block-removing knowledge distillation,
simple, but effective.
2.  This work demonstrates the effectiveness of their proposed method on different model sizes, different tasks, including text-image generation and personalized text-image generation.
The paper provides robust evidence of the approach's versatility across varying model sizes and tasks, which include both text-image and personalized text-image generation.


**Weaknesses:**

1. The main concern lies in the hand-crafted block removal process from the original DDPM. It may be more reasonable to calculate an importance score for each block and remove them accordingly. The simplicity of the current approach, while appreciated, seems to compromise on the degree of novelty. Please further clarify this.

2. A comparison of different sampling steps is missing. Will the model compression influence the trade-off between fidelity and sampling efficiency?


3. The proposed methods seem to be general for the diffusion model.  Could the authors elaborate on the reason not to include experiments on general image generation?

**Questions:**

My key questions stem from innovation in weakness. If it is solved, I am willing to improve my score.

**Limitations:**

Yes

---

> ### Author Rebuttal · Authors · 2023-08-06
>
> Thank you very much for your valuable feedback. By incorporating the experiments you recommended, we believe our paper has been strengthened. We kindly ask you to reconsider your rating in light of our responses and the newly conducted experiments. If you have any further questions or concerns, please let us know.
>
> > Importance score for each block and group in SD
>
> * We sincerely appreciate your valuable comment. We newly analyze the “block-wise sensitivity” and “block-groupwise sensitivity” (see **PDF-Fig. R1** and the below table). The results are aligned with our architectural choices (e.g., removing inner blocks and first RA pairs in the down stages), clearly supporting our study. We remark that this is the first to perform pruning sensitivity analyses for large diffusion models.
> * We note that block removal is not a viable action for all the blocks, because of their different channel dimensions of input and output (particularly for the residual blocks in the later up stages). To handle this, we replace them with channel interpolation modules to mimic the removal while retaining the information.
> * Additionally, treating a ResBlock (R) and an AttentionBlock (A) as a pair and an integrated unit could be a valuable approach, given their complementary roles in processing different information types (R for time embeddings and A for text information) and the design philosophy of the original model.
>
> A portion of the block-groupwise sensitivity (please see **PDF-Fig. R1** for the block-wise sensitivity and the full groupwise analysis)
>
> | Stage | Removed/Replaced* |  FID↓ |  IS↑ | CLIP Score↑ |
> |-------|:-----------------:|:-----:|:----:|:-----------:|
> | Down  |      RA(1,1)      |  46.4 | 22.6 |    0.240    |
> | Down  |      RA(1,2)      |  35.2 | 28.2 |    0.263    |
> | Down  |      RA(2,1)*     | 226.4 |  5.1 |    0.043    |
> | Down  |      RA(2,2)      |  30.2 | 29.3 |    0.264    |
> | Down  |      RA(3,1)*     | 107.5 | 12.3 |    0.092    |
> | Down  |      RA(3,2)      |  19.0 | 32.4 |    0.286    |
> | Down  |      RR(4,1)      |  19.9 | 35.2 |    0.297    |
> | Mid   |      RAR(5,1)     |  22.1 | 31.9 |    0.264    |
> | Up    |     RRR(6,1)*     |  22.1 | 31.8 |    0.261    |
> | Up    |      RA(7,1)*     | 103.0 | 10.9 |    0.089    |
> | Up    |      RA(7,2)*     | 135.6 |  8.1 |    0.041    |
> | Up    |      RA(7,3)*     | 132.9 |  6.9 |    0.041    |
> | ...   |      ...    | ... |  ... |    ...    |
> | -     |  Original SD-v1.4 |  19.9 | 35.3 |    0.300    |
>
>
> * We further analyze different architectures by removing less important blocks in **PDF-Fig. R1(a)**. Specifically, the CLIP scores are sorted and used to remove the top 14 blocks (with CLIP score > 0.29), 21 blocks (CLIP score > 0.265), 24 blocks (CLIP score > 0.245 and FID < 25), and 26 blocks (CLIP score > 0.20 and FID < 100). The post-pruning results are as follows, while the exploration of training these architectures remains an interesting future direction. While it remains uncertain whether extensively eliminating the majority of attention blocks (which handle text embeddings) can result in good text-aligned images, these could serve as valuable baselines for our models. Thank you once again for your comment.
>
> Post-pruning results (without training) on MS-COCO 512×512 5K
>
> | Architecture     | # Params, Whole (U-Net) | FID↓  | IS↑ | CLIP Score↑ |
> |------------------|-------------------------|-------|-----|-------------|
> | Remove 14 blocks | 781M (609M)             | 135.1 | 6.9 | 0.1226      |
> | Remove 21 blocks | 604M (431M)             | 274.6 | 2.0 | 0.0115      |
> | Remove 24 blocks | 456M (284M)             | 274.9 | 2.0 | 0.0120      |
> | Remove 26 blocks | 452M (279M)             | 348.6 | 1.2 | 0.0797      |
> | BK-SDM-Base      | 752M (580M)             | 537.9 | 2.1 | 0.0012      |
> | BK-SDM-Small     | 655M (483M)             | 539.4 | 2.1 | 0.0009      |
> | BK-SDM-Tiny      | 496M (324M)             | 537.4 | 2.2 | 0.0007      |
>
>
> >Different sampling steps
>
> Thank you for your insightful comment. We present additional results by varying the number of sampling steps on MS-COCO 512×512 5K. Please find **PDF-Fig. R3** and the below tables, where BK-SDM-Base (0.76B) has fewer parameters than the original SDM (1.04B). Compared to the baseline trained only with the denoising task loss, our distillation pretraining achieves much better trade-off between generation quality and sampling efficiency.
>
> For FID↓ (Please refer to PDF-Fig. R3 for IS↑)
>
> | # Denoising Steps  | 10    | 15    | 25    | 40    |
> |--------------------|-------|-------|-------|-------|
> | BK-Base, No KD   | 60.53 | 34.33 | 30.68 | 30.41 |
> | BK-Base, With KD | 26.65 | 20.95 | 22.39 | 22.66 |
> | Original SDM-v1.4   | 21.34 | 17.47 | 19.92 | 20.16 |
>
> For CLIP Score↑
>
> | # Denoising Steps  | 10     | 15     | 25     | 40     |
> |--------------------|--------|--------|--------|--------|
> | BK-Base, No KD   | 0.1905 | 0.2290 | 0.2431 | 0.2444 |
> | BK-Base, With KD | 0.2633 | 0.2851 | 0.2909 | 0.2924 |
> | Original SDM-v1.4   | 0.2783 | 0.2954 | 0.2999 | 0.3016 |
>
> >Extention to general image generation
>
> The main reason for our focus on SD is its substantial compute demands, despite its growing popularity as backbones for vision-related applications. However, agreeing with your comment, we further leverage our method for general image generation. In **PDF-Fig. R4**, we compress a latent diffusion model for unconditional generation with the CelebA face dataset (i.e., CompVis/ldm-celebahq-256) with the same approach in BK-SDM-Small. The results clearly show the architectural applicability of our model and the importance of KD pretraining.

---

> > ### Comment · Reviewer_WEDt · 2023-08-16
> > **Thanks for the response**
> >
> > I sincerely appreciate the rebuttal from the author. However, the response to my first question was mistaken. Could you set a metric to measure the importance of each block, such as redundancy between channels or different blocks? This would guide the decision on which block should be removed. If the removed block is selected based on experimental results, this approach seems more akin to a technical report than an innovative contribution, despite the model being a diffusion model. For instance, if we alter the backbone for the diffusion model, conducting experiments for each removed block would be required again, making the process time-consuming and lacking generalizability. Based on the experimental efforts, I can only raise my score to 5 as encouragement. However, if other reviewers feel that further improvements are needed and decide to borderline reject this work, I also agree with it.

---

> > > ### Author Response · Authors · 2023-08-16
> > > **Thank you,**
> > >
> > > Dear Reviewer WEDt,
> > >
> > > Thank you very much for your feedback and increased rating.
> > >
> > > We would like to clarify the process of sensitivity analysis: when removing/replacing each block, we observed how generation scores vary. If a block played a crucial role, removing it causes a significant degradation in the generation score. In our experience, the final scores serve as the most indicative metrics for assessing the importance of blocks.
> > >
> > > Thank you for sharing your concern; we understand your point. However, we would like to clarify that while the sensitivity analysis supports our architectural choices, it was not initially incorporated into the original design. The design was formulated based on human knowledge (prioritizing blocks with altered channel dimensions) and was substantiated through empirical validation. Furthermore, it was aligned with previous studies involving DistilBERT [NeurIPS Workshop, 2019] and Cut Inner Layers [ICML Workshop, 2022]. We would like to emphasize the growing significance of empirical contributions for large foundational models. To the best of our knowledge, prior compression studies on CNNs and Transformers have not addressed such extensive models.
> > >
> > > In addition, our work can be applied to other SD versions within the range of v1.1 to v1.5 using the same architecture, and to SDM-v2 with a similarly designed architecture (involving RA pairs and U-Net skip connections). We kindly ask for your understanding that the Stable Diffusion family stands as one of the rarely open-sourced foundational models for text-to-image synthesis, which has triggered a paradigm shift in both research and industrial communities. Moreover, the architectural design of SD's U-Net is widely utilized; thanks to your comment, we have already demonstrated the versatility of our approach across different LDMs.
> > >
> > > Beyond the architectural aspects, our work also unlocks the potential of Feature KD for compressing diffusion models and demonstrates the usage in personalized generation.
> > >
> > > We hope our responses convince you about the value of our approach. We express our thanks once again for your valuable insights and your dedicated participation in this review process.

---

> > > > ### Comment · Reviewer_WEDt · 2023-08-16
> > > > **Thanks for the response**
> > > >
> > > > Thanks for your clarification. If possible, I expect you can add the detailed methods in the revision. I cannot find this description that is vital in your paper. It looks that this only exists in the response.

---

> > > > > ### Author Response · Authors · 2023-08-16
> > > > >
> > > > > Dear Reviewer WEDt,
> > > > >
> > > > > Thanks for your prompt response. We apologize for any inconvenience caused by including the method description in the caption of PDF-Fig. R1 (i.e., "we remove each block to examine its effect on generation performance.")
> > > > >
> > > > > Following your advice, we will clarify it in our global response and further author-reviewer discussions. We will also include the method details and results regarding the sensitivity analysis in our updated manuscript.
> > > > >
> > > > > Best regards,
> > > > >
> > > > > Authors

---

> > > > > > ### Comment · Reviewer_WEDt · 2023-08-18
> > > > > >
> > > > > > Thanks. I have no other questions.

---

> > > > > > > ### Author Response · Authors · 2023-08-19
> > > > > > >
> > > > > > > Thank you for your confirmation.
> > > > > > >
> > > > > > > Best regards,
> > > > > > >
> > > > > > > Authors

---

### Author Rebuttal · Authors · 2023-08-06

We would like to thank the reviewers for their constructive feedback, which has enabled us to make improvements to the paper.

> Background of this study:

Although Stable Diffusion (SD) has received significant attention from both research and industry communities, the substantial computation and data for its training (e.g., **256 A100 GPUs and 600M LAION pairs** for SD-v1.4) are unfeasible for most researchers.

We also note that previous compression studies on CNNs and Transformers have not dealt with such large foundation models: e.g., ResNet152 (with 60.2M parameters) and ViT_L_16 (306M params) are much smaller than SD-v1.4 (1033M params). Moreover, U-Net architectures are arguably more complex due to the necessity of considering skip connections across the network, making the structural block removal inside them not straightforward. Furthermore, prior studies using distillation for fewer steps towards efficient diffusion are highly valuable but cannot reduce the model size (because of distilling over identically structured models), which our architectural compression can address.


> Importance of this study:


We are delighted that the reviewers recognize the importance of our study, which addresses the architectural compression of SD as a pioneering direction [Reviewers M23o, Gqvg], successfully achieves Compressed SD with very limited resources (**1 A100 GPU and 0.22M pairs**) which individuals can access [WEDt, M23o, Gqvg, wHDk], and shows the capability as lightweight backbones for personalized generation [WEDt, ZGLR]. Our comprehensive analyses and ablation studies (useful to the community) have drawn the reviewers' interest  [WEDt, ZGLR, Gqvg].


>New experiments per reviewers’ suggestion: — please find **Figure_Rebuttal.pdf** (attached below)

To the best of our knowledge, (1) and (2) are the first analyses about the importance of each block and group in Stable Diffusion. They are aligned with our architectural choice, clearly supporting our study.

* (1) Analyzing the block-wise sensitivity — Fig. R1(a) in Figure-PDF
* (2) Analyzing the block-groupwise sensitivity — Fig. R1(b)
* (3) Varying the pretraining data size from 212K pairs to {2256K, 1128K, 100K, 50K, 11K} pairs — Fig. R2
* (4) Varying the number of denosing steps from 25 to {10, 15, 40} — Fig. R3
* (5) Applying to Latent Diffusion for unconditional image generation — Fig. R4
* (6) Further comparison to the filtered-data GLIDE — Fig. R5

>Architectural variations

Please find another architecture **BK-SDM-Tiny** (obtained with further inner-stage removal) in Supplementary Section B, which showcases an impressive compression rate (**51% fewer parameters**, 43% latency improvement).
* We believe we have extensively studied the effect of different compression rates by introducing BK-SDM-{Base, Small, Tiny} with {27%, 36%, 51%} reduced model size.

Additionally, through this rebuttal, we newly introduce other structural variations by eliminating less important blocks based on the block-wise sensitivity analysis (PDF-Fig. R1(a)).

---

> ### Author Response · Authors · 2023-08-16
> **Block-level sensitivity analyses**
>
> Thanks to the comment from Reviewer WEDt, we would like to provide further clarification on the process of sensitivity analyses (1) and (2):
> - During the process of removing/replacing each block, we observed the variations in generation scores.
> - In cases where a block played a crucial role, its removal led to a significant degradation in the generation score.
>
> In our experience, the final scores serve as the most indicative metrics for assessing the importance of blocks.

---

### Author Response · Authors · 2023-08-14
**A Gentle Reminder for Author-Reviewer Discussions**

Dear Reviewers,

Thank you for your time and efforts in reviewing our paper. We would greatly appreciate if you would provide us with feedback on our rebuttal. We have tried to address all your comments and are open to participating in fruitful discussions.

Thank you very much,

Authors

---

### Comment · Reviewer_Gqvg · 2023-08-16

I am Reviewer Gqvg. I was initially positive about this paper and remain positive about this paper, even after reading the more-negative feedback from the other reviewers.

The general summary of the concerns of the reviewers seem to hinge on whether this paper has enough _technical novelty_ to merit an acceptance to the conference. In my personal view, technical novelty in the sense of having to produce some arbitrary (and often complex for the sake of novelty) algorithm should not be a requirement for a paper to be accepted. I see this often in systems papers where authors may produce some well-designed system (for example, a mechanism to let you very easily tune parameters that are otherwise hard to parameterize because those parameters actually define something like window sizes for writing image processing kernels) but need to also propose some complicated search algorithm for those parameters for the sake of 'novelty'. In reality, what really made those papers valuable in retrospect was the 'simple' contribution and not the complicated search algorithm.

I see this paper in the same way. The authors provide an incremental, but still useful measurement study (especially with the additional block-wise sensitivity analysis they provide) that will guide practitioners who work in enabling text-to-image generation on more resource constrained settings. While incremental, this is an impactful analysis because there is an immediate real-world need where even a 50% decrease in parameters can significantly increase the number of legacy GPUs that people can run text-to-image generation on. This is also something that is currently performed in ad-hoc fashion and hidden in the depths of Discord servers of text-to-image gurus without real evidence or structured analysis; making this analysis instead available to the wider community and in the open is a definite need. This paper will also be sure to generated follow-on works that will use the results of the analysis they provide, and either 1. come up with some theoretical explanations that lead in the sensitivities they see or 2. an algorithm that can provide a general pruning mechanism. I don't think this paper necessarily needs to provide all of these things at once.

The authors clearly put in a lot of effort into the rebuttal, so it would be great if the other reviewers can provide their feedback based on this rebuttal as well.

---

> ### Author Response · Authors · 2023-08-17
> **Thank you,**
>
> Dear Reviewer Gqvg,
>
> We greatly appreciate your recognition of our work and the thoughtful perspective you've shared. Your support means a lot to us.
>
> We are grateful for all your time and effort in reviewing our work and participating in this review process.
>
> Sincerely,
>
> Authors

---

### Author Response · Authors · 2023-08-21

We deeply thank the reviewers for their time, constructive feedbacks, and responses to the rebuttal.

We believe that empirical contributions become important in the era of large foundational models. Our study, which *proposes appropriate compressed architectures for Stable Diffusion, uncovers the potential of feature KD, presents valuable analyses (#), and makes experiments feasible for individual researchers*, could benefit the community. We would greatly appreciate it if you could take these aspects into consideration.
- (#) Our work includes the MS-COCO benchmark, ablation study, performance change over training progress, effect of classifier-free guidance scales, effect of the number of sampling steps, effect of training data volume, block-level sensitivity analyses, and application to unconditional image generation.

Sincerely,

Authors

---

### Decision · Program_Chairs · 2023-09-21

**Decision:**

Reject

**Comment:**

The paper received mixed reviews: Reviewers acknowledge the paper for its practical approach to reducing computational overhead, and the effort to run thorough analysis. However, there are serious concerns about the lack of methodological novelty, and the absence of an in-depth analysis of the trade-off between speed and quality. The rebuttal addresses some of the questions but does not fully satisfy concerns about the paper's novelty. After a discussion among ACs, SACs, and PCs, a consensus was reached to not accept the paper, majorly due to the novelty concerns. Despite this decision, we all recognize its practical implications and the thoroughness of its experiments.